

# Estimation of rate coefficients and branching ratios for gas-phase reactions of OH with aliphatic organic compounds for use in automated mechanism construction

Michael E. Jenkin[1], Richard Valorso[2], Bernard Aumont[2], Andrew R. Rickard[3,4], Timothy J. Wallington[5]

[1] Atmospheric Chemistry Services, Okehampton, Devon, EX20 4QB, UK
[2] LISA, UMR CNRS 7583, Université Paris Est Créteil et Université Paris Diderot, 94010 Créteil, France
[3] Wolfson Atmospheric Chemistry Laboratories, Department of Chemistry, University of York, York, YO10 5DD, UK
[4] National Centre for Atmospheric Science, University of York, York, YO10 5DD, UK
[5] Research and Advanced Engineering, Ford Motor Company, SRL-3083, PO Box 2053, Dearborn, MI 48121-2053, USA

*Correspondence to*: Michael E. Jenkin (atmos.chem@btinternet.com)

**Abstract.** Reaction with the hydroxyl (OH) radical is the dominant removal process for volatile organic compounds (VOCs) in the atmosphere. Rate coefficients for reactions of OH with VOCs are therefore essential parameters for chemical mechanisms used in chemistry-transport models, and are required more generally for impact assessments involving estimation of atmospheric lifetimes or oxidation rates for VOCs. Updated and extended structure-activity relationship (SAR) methods are presented for the reactions of OH with aliphatic organic compounds, with the reactions of aromatic organic compounds considered in a companion paper. The methods are optimized using a preferred set of data including reactions of OH with 489 aliphatic hydrocarbons and oxygenated organic compounds. In each case, the rate coefficient is defined in terms of a summation of partial rate coefficients for H abstraction or OH addition at each relevant site in the given organic compound, so that the attack distribution is defined. The information can therefore guide the representation of the OH reactions in the next generation of explicit detailed chemical mechanisms. Rules governing the representation of the subsequent reactions of the product radicals under tropospheric conditions are also summarized, specifically their reactions with $O_2$ and competing processes.

## 1 Introduction

It is well documented that volatile organic compounds (VOCs) are emitted into the atmosphere in substantial quantities from both anthropogenic and biogenic sources (e.g. Guenther et al., 2012; Huang et al., 2017). The degradation of VOCs has a major influence on the chemistry of the troposphere, contributing to the formation of ozone ($O_3$), secondary organic aerosol (SOA) and other secondary pollutants (e.g. Haagen-Smit and Fox, 1954; Went, 1960; Andreae and Crutzen, 1997; Jenkin and Clemitshaw, 2000; Hallquist et al., 2009).



The complete gas-phase oxidation of emitted hydrocarbons and oxygenated organic compounds into carbon dioxide and water proceeds via highly detailed mechanisms, and produces a wide variety of intermediate oxidized organic products (e.g. Saunders et al., 2003; Aumont et al., 2005). As a result of the complexity of the emitted speciation, and of the degradation chemistry, the atmosphere contains an extremely large number of structurally different organic compounds, which possess a

wide range of reactivities. For the majority of these, reaction with the hydroxyl (OH) radical is the dominant or exclusive removal process, such that it plays an important role in determining the atmospheric lifetime and impact of a given organic compound. As a result, quantified rate coefficients for the reactions of OH with organic compounds are essential parameters for chemical mechanisms used in chemistry-transport models, and are invariably required more generally for environmental assessments of their impacts, e.g. to estimate the kinetic component of ozone formation potentials (Bufalini et al., 1976;

Carter, 1994; Derwent et al., 1998; Jenkin et al., 2017) or atmospheric lifetimes for the calculation of global warming potentials (e.g. Kurylo and Orkin, 2003). In addition to the total rate coefficient, quantification of the branching ratio for attack of OH at each site within a given compound is required for explicit representation of the subsequent oxidation pathways in chemical mechanisms.

As part of the present work, a set of preferred kinetic data has been assembled for the reactions of OH with 556 organic

compounds, based on reported experimental studies, of which 489 are for aliphatics (see Sect. 2 for further details). Previous assessments using explicit organic degradation mechanisms have demonstrated that the atmosphere contains an almost limitless number of organic compounds (e.g. Aumont et al., 2005), for which it is impractical to carry out experimental kinetics studies. This has resulted in the development of estimation methods for OH rate coefficients (e.g. see Calvert et al., 2015; and references therein), which have been applied widely in chemical mechanisms and impact assessments.

In the present paper, updated structure-activity relationship (SAR) methods are presented for the reactions of OH with aliphatic organic compounds, with the reactions of aromatic organic compounds considered in a companion paper (Jenkin et al., 2018a). In each case, the rate coefficient is defined in terms of a summation of partial rate coefficients for H atom abstraction or OH addition at each relevant site in the given organic compound, so that the attack distribution is also defined. Particular use is made of the methods reported by Kwok and Atkinson (1995) and Peeters et al. (2007), which are updated

and extended on the basis of the current preferred data. These approaches are also supplemented by newly developed methods for some compound classes (e.g. cumulative dienes and alkynes), and application of the methods is illustrated with examples in the Supplement.

The information is currently being used to guide the representation of the OH-initiation reactions in the next generation of explicit detailed chemical mechanisms, based on the Generator for Explicit Chemistry and Kinetics of Organics in the

Atmosphere, GECKO-A (Aumont et al., 2005), and the Master Chemical Mechanism, MCM (Saunders et al., 2003). It therefore contributes to a revised and updated set of rules that can be used in automated mechanism construction, and provides formal documentation of the methods. To facilitate this, rules governing the representation of the initial rapid reactions of the product radicals under tropospheric conditions are also summarized, specifically their reactions with $O_2$ and





competing processes. The treatment of the subsequent chemistry (e.g. reactions of peroxy radicals) will be reported elsewhere (e.g. Jenkin et al., 2018b).

## 2 Preferred kinetic data

A set of preferred kinetic data has been assembled from which to develop and validate the estimation methods for the OH rate coefficients. The complete set includes data for 172 hydrocarbons and 384 oxygenated organic compounds. The subset relevant to the present paper includes 298 K data for a total of 489 organic compounds, comprising alkanes (49 reactions), alkenes/polyalkenes (92 reactions), alkynes (6 reactions), saturated oxygenated organic compounds (259 reactions) and unsaturated aliphatic oxygenated organic compounds (83 reactions), with temperature dependences also defined for a subset of 153 organic compounds. In two cases, the preferred rate coefficient is an upper limit value, and in one case a lower limit value. The information is provided as a part of the Supplement (spreadsheets SI_1 to SI_5). As described in more detail in Sects. 3.2 and 4.2, the oxygenates include both monofunctional and multifunctional compounds containing a variety of functional groups that are prevalent in both emitted VOCs and their degradation products, namely -OH, -OOH, -C(=O)-, -O-, -C(=O)O-, -ONO$_2$, -NO$_2$ and -C(=O)OONO$_2$. For a core set of 73 reactions, preferred kinetic data are based on the evaluations of the IUPAC Task Group on Atmospheric Chemical Kinetic Data Evaluation (http://iupac.pole-ether.fr/). The remaining values are informed by recommendations from other key evaluations with complementary coverage (e.g. Atkinson and Arey, 2003; Calvert et al., 2008, 2011), and have been revised and expanded following review and evaluation of additional data not included in those studies (as identified in spreadsheets SI_1 to SI_5).

## 3 Saturated organic compounds

The reactions of OH with saturated organic compounds result almost exclusively in the abstraction of an H-atom from a C-H or O-H bond. The representation of H-atom abstraction reactions in the current methodology is an update and extension to the widely-applied SAR method of Kwok and Atkinson (1995), for which selected updated parameters for 298 K have also been reported in some other more recent studies (e.g. Atkinson, 2000; Bethel et al., 2001; Calvert et al., 2008, 2011). The estimated rate coefficients are thus based on a summation of rate coefficients for H-atom abstraction from the primary (-CH$_3$), secondary (-CH$_2$-) and tertiary (-CH<) groups, and from any hydroxy (-OH) and hydroperoxy (-OOH) groups in the given organic compound, which are calculated as follows:

$$k(\mathrm{CH_3\text{-}X}) = k_{\mathrm{prim}}\ \mathrm{F(X)} \tag{1}$$

$$k(\mathrm{X\text{-}CH_2\text{-}Y}) = k_{\mathrm{sec}}\ \mathrm{F(X)\ F(Y)} \tag{2}$$

$$k(\mathrm{X\text{-}CH(\text{-}Y)\text{-}Z}) = k_{\mathrm{tert}}\ \mathrm{F(X)\ F(Y)\ F(Z)} \tag{3}$$

$$k(\text{-OH}) = k_{\mathrm{abs(\text{-}OH)}} \tag{4}$$



$k(\text{-OOH}) = k_{\text{abs(-OOH)}}$ (5)

$k_{\text{prim}}$, $k_{\text{sec}}$ and $k_{\text{tert}}$ are the respective group rate coefficients for abstraction from primary, secondary and tertiary groups for a reference substituent; and F(X), F(Y) and F(Z) are factors that account for the effects of the substituents X, Y and Z. The reference substituent is defined as "-CH$_3$", such that F(-CH$_3$) = 1.00 (Atkinson, 1987; Kwok and Atkinson, 1995). $k_{\text{abs(-OH)}}$

and $k_{\text{abs(-OOH)}}$ are the rate coefficients for H-atom abstraction from -OH and -OOH groups, the values of which are assumed to be independent of the identity of neighbouring substituent groups, as also previously assumed for $k_{\text{abs(-OH)}}$ by Kwok and Atkinson (1995).

A number of studies, including Kwok and Atkinson (1995), have defined rate coefficients for reaction at other specific oxygenated groups, with these also being assumed to be independent of the identity of neighbouring substituent groups.

These include abstraction of the H atom in carboxyl (-C(=O)OH) groups (Kwok and Atkinson, 1995), and also H-atom abstraction from the formyl group in formate esters (Le Calvé et al., 1997; Calvert et al., 2015). In the present work, it was found that the performance of the method for some further organic oxygenates could be improved by assigning fixed rate coefficients for H-atom abstraction from C-H bonds in specific environments, i.e. from formyl groups in aldehydes, and adjacent to -O- linkages in ethers and diethers. This is discussed further in Sect. 3.2.

**3.1 Alkanes**

**3.1.1 Acyclic alkanes**

The values of $k_{\text{prim}}$, $k_{\text{sec}}$ and $k_{\text{tert}}$, and of the substituent factors F(-CH$_2$-), F(-CH<) and F(>C<), were initially optimized for 298 K, using the preferred kinetic data for acyclic (non-methane) alkanes in the dataset, which comprise 12 linear (*n*-) alkanes and 14 branched alkanes. This was achieved by minimizing the summed square deviation, $\Sigma((k_{\text{calc}}-k_{\text{obs}})/k_{\text{obs}})^2$, where $k_{\text{obs}}$ is the preferred

(observed) value of the rate coefficient, and $k_{\text{calc}}$ is the value calculated using the SAR. As in previous studies (Kwok and Atkinson, 1995; Calvert et al., 2008), it was found that there was little benefit in using independent values of F(-CH$_2$-), F(-CH<) and F(>C<), and a single value of F(-CH$_2$-, -CH<, >C<) was therefore optimized for simplicity. The resultant values of the optimized parameters are given in Tables 1 and 2, and a correlation of $k_{\text{calc}}$ and $k_{\text{obs}}$ at 298 K is shown in the upper panel of Fig. 1. The updated method results in a value of $\Sigma((k_{\text{calc}}-k_{\text{obs}})/k_{\text{obs}})^2$ that is lower than those obtained by using the parameters optimized

to earlier datasets by Kwok and Atkinson (1995) and Calvert et al. (2008), by factors of 1.6 and 1.1, respectively.

Temperature-dependent recommendations are available for 17 acyclic (non-methane) alkanes in Arrhenius format ($k$ = A exp(-($E/R$)/T), as given in the preferred data in the Supplement (spreadsheet SI_1). These were used to provide optimized values of the temperature coefficient ($E/R$) and pre-exponential factor (A) for the group rate coefficients, $k_{\text{prim}}$, $k_{\text{sec}}$ and $k_{\text{tert}}$ (see Table 1), with the assumption that the weak temperature dependence of the substituent factor F(-CH$_2$-, -CH<, >C<) can be expressed as A$_{\text{F(X)}}$

exp(-B$_{\text{F(X)}}$/T), with A$_{\text{F(X)}}$ = 1.00, as applied previously by Atkinson (1987) and Kwok and Atkinson (1995) (see Table 2). Optimization was achieved by calculating values of $k_{\text{calc}}$ at even 1/T intervals over the recommended temperature range for each alkane (with an imposed upper limit of 400 K, where applicable), and determining a composite $(E/R)_{\text{calc}}$ value from a least squares





linear regression of the data on an Arrhenius (i.e. ln(k) vs. 1/T) plot. The values of $(E/R)_{prim}$, $(E/R)_{sec}$ and $(E/R)_{tert}$ were varied to minimize the summed square deviation in the composite temperature coefficients, $\Sigma((E/R)_{calc}-(E/R)_{obs})^2$. The resultant $(E/R)_{calc}$ values are compared with the recommended $(E/R)_{obs}$ values in the lower panel of Fig. 1 (see also, Fig. S1 in the Supplement). The values of $A_{prim}$, $A_{sec}$ and $A_{tert}$ were automatically returned from the corresponding optimized $E/R$ and $k_{298K}$ values. The resultant

trend in $(E/R)_{prim}$, $(E/R)_{sec}$ and $(E/R)_{tert}$ values (Table 1) shows a logical progression, indicative of the progressive weakening of the C-H bond in primary, secondary and tertiary groups.

### 3.1.2 Cyclic alkanes

The parameter values determined above can also be applied to calculate rate coefficients for the reactions of OH with cyclic alkanes. As discussed previously by Kwok and Atkinson (1995), ring-strain has an impact on the H-atom abstraction kinetics in

cyclic systems. The data for 22 cyclic alkanes were therefore used to optimize empirical ring-strain factors, $F_{ring}$, for 3-member through to 8-member rings, leading to the values given in Table 3. These values need to be applied in conjunction with the neighbouring group ($F(X)$) factors, such that the following equations apply to the calculation of H-atom abstraction rate coefficients from intra-cyclic "-$CH_2$-" and "-CH<" groups in monocyclic alkanes:

$$k(X\text{-}CH_2\text{-}Y) = k_{sec}\,F(X)\,F(Y)\,F_{ring} \qquad (6)$$

$$k(X\text{-}CH(\text{-}Y)\text{-}Z) = k_{tert}\,F(X)\,F(Y)\,F(Z)\,F_{ring} \qquad (7)$$

For polycyclic alkanes, a value of $F_{ring}$ needs to be applied for each ring for which the given "-$CH_2$-" or "-CH<" group is a component, as discussed by Kwok and Atkinson (1995).

Similarly to the values derived (or assumed) by Kwok and Atkinson (1995), the optimized $F_{ring}$ values for 6-, 7- and 8-member rings are close to unity, with a progressive decrease in the values for the smaller more strained rings. Although a six-member ring

is a classical example of a strain-free system (e.g. Calvert et al., 2008), the recommended data for most alkanes with six-membered rings suggest a slight deactivating effect relative to acyclic "-$CH_2$-" groups, particularly in the case of cyclohexane itself. A correlation of the optimized 298 K values of $k_{calc}$ and $k_{obs}$ is shown in the upper panel of Fig. 1.

Temperature-dependent parameters are recommended for the series of unsubstituted monocyclic alkanes, cyclopropane through to cyclooctane, in Arrhenius format (see spreadsheet SI_1). The recommended $E/R$ values for the larger systems (cyclohexane,

cycloheptane and cyclooctane) are similar to those derived from the overall temperature coefficient $k_{sec}\,F(\text{-}CH_2\text{-})^2$ (= 377 K), derived above for H-atom abstraction from "-$CH_2$-" groups in long chain acyclic alkanes. This is compatible with $F_{ring}$ having no significant temperature dependence for 6-, 7- or 8-membered rings, and also consistent with their near-unity 298 K values. In the cases of cyclopentane, cyclobutane and, particularly, cyclopropane, the recommended $E/R$ values are progressively more elevated (450 K, 510 K and 1300 K, respectively), and it was necessary to assign temperature-dependent values of $F_{ring} = A_{F(ring)}\exp(-$

$B_{F(ring)}/T)$ for 3-, 4- and 5-member rings, as shown in Table 3. In these cases, the values of $B_{F(ring)}$ were once again varied to minimize the summed square deviation in the composite temperature coefficients, with values of $A_{F(ring)}$ automatically returned



from the procedure. The resultant calculated values of $E/R$ are compared with the recommended values in the lower panel of Fig. 1 (see also, Fig. S1).

## 3.2 Saturated organic oxygenates

### 3.2.1 Compounds containing carbonyl and hydroxyl groups

Consistent with the approach adopted by Kwok and Atkinson (1995), the value of the rate coefficient for H-atom abstraction from a hydroxy group, $k_{abs(-OH)}$, is based on the rate coefficient for abstraction from the -OH group in methanol, as recommended by the IUPAC panel (see Table 4). The values of a number of other parameters, shown in Tables 5 and 6, were optimized using the preferred data for compounds containing combinations of carbonyl and hydroxy groups. These included 33 alcohols and diols, 22 aldehydes, 17 ketones, 6 dicarbonyls, 8 hydroxyaldehydes and 18 hydroxyketones. In the original method of Kwok and

Atkinson (1995), abstraction of the H-atom from the formyl group in aldehydes was logically represented by defining a substituent factor, F(=O), which was used in combination with $k_{tert}$ and any other relevant substituent factors. In conjunction with the updates to substituent effects for hydroxy groups reported subsequently by Bethel et al. (2001), the method has been shown to provide a poor representation of the rate coefficients for hydroxyaldehydes, with overestimates of up to a factor of four relative to the observed values (Baker et al., 2004; Mason et al., 2010; Calvert et al., 2011). This suggests that hydroxy groups have a

significant deactivating effect on abstraction from formyl groups, whereas they generally activate abstraction from alkyl groups (Bethel et al., 2001). In conjunction with the observed increasing trend in $k_{298K}$ with increasing alkyl substitution in the organic group, it appears that the reactivity of the formyl group is influenced by the inductive effect of the organic group.

In the present work, the performance of the method is significantly improved by defining a set of rate coefficients for H-atom abstraction from formyl groups in RC(=O)H, for a variety of different classes of R. These are shown in Table 4, and are applied

independently of substituent factors. The displayed parameters ($k_{abs(-CHO)n}$ etc.) are generic and apply to the series of classes of R identified and described in Table 4. They are also used as default rate coefficients for additional classes containing substituents for which there are currently no data (see Table 4 notes). The values of the other (un-named) rate coefficients in Table 4 relate only to specific compounds, and are included to illustrate trends of increasing substitution in R.

The parameters in Table 4 were optimized in conjunction with the substituent factors listed in Table 5, which relate to the general

influence of hydroxyl and carbonyl groups on H abstraction from sites other than formyl groups in these compounds. The parameter values were initially optimized for 298 K, using a global fit to the preferred kinetic data indicated above, using the values of $k_{prim}$, $k_{sec}$ and $k_{tert}$ (and values of $F_{ring}$ reported for cycloalkanes in Table 3). Consistent with the approach in previous studies, the substituent factors describe the effects of α- or β- carbonyl groups (Kwok and Atkinson, 1995) and of α- or β- hydroxy groups (Bethel et al., 2001). The resultant values of the optimized parameters are given in Tables 4 and 5, and a

correlation of $k_{calc}$ and $k_{obs}$ at 298 K is shown in Fig. 2. As a result of the inclusion of the effects of β- groups in determining $k_{calc}$, there are instances where the neighbouring group substituent factor, F(X), is influenced by two groups ($X_1$ and $X_2$), such that a combination of $F(X_1)$ and $F(X_2)$ needs to be applied. For the present set of data, this occurs for 9 compounds where





one or more sites is influenced by both a β- carbonyl and a β- hydroxy group as part of the same substituent, e.g. containing a -CH(OH)(C(=O)-)- sub-structure. In these cases, it was found that including the associated activating effect of both groups (i.e. $F(X) = F(X_1).(FX_2)$) resulted in systematic overestimation of the rate coefficients, whereas the data were generally well described if the assumption, $F(X) = (F(X_1)(FX_2))^{\frac{1}{2}}$, was applied. Where relevant, this approach was therefore adopted throughout the present work for H-atom abstraction reactions.

The estimation method reproduces the observed 298 K values to within a factor of two for almost all of the compounds considered, with particularly good descriptions for aldehydes (within 30 %) and hydroxyaldehydes (within 10 %) due in part to the adjusted methodology described above. Similarly to the results of Bethel et al. (2001) and Mason et al. (2010), the method systematically underestimates the rate coefficients for 1,3- and 1,4-di-alcohols, by factors in the range 1.7 - 2.5. As also discussed previously by Calvert et al. (2011), this is likely due to longer range influences of hydroxy substituents that are difficult to include in a practical SAR method.

Temperature-dependent recommendations are available for 32 compounds containing combinations of carbonyl and hydroxy groups (in addition to formaldehyde and glyoxal). These were used to provide representative values of the temperature coefficient ($E/R$) and pre-exponential factor (A) for the group rate coefficients given in Table 4, and for the temperature coefficient ($B_{F(X)}$) and pre-exponential factor ($A_{F(X)}$) for the substituent factors given in Table 5. The values of ($E/R$) for the group rate coefficients and $B_{F(X)}$ for the substituent factors were varied with the aim of minimizing the summed square deviation in the composite temperature coefficients, $\Sigma((E/R)_{calc}-(E/R)_{obs})^2$, for the contributing set of compounds. The resultant $(E/R)_{calc}$ values are compared with the recommended $(E/R)_{obs}$ values in the lower panel of Fig. 2 (see also, Fig. S2). The values of A were automatically returned from the corresponding optimized $E/R$ and $k_{298K}$ values, and $A_{F(X)}$ from the corresponding optimized $B_{F(X)}$ and $F(X)_{298K}$ values.

The preferred data also include rate coefficients for seven cycloketones (specifically defined as compounds where the >C=O group forms part of a cycle). These were not included in the optimization procedure described above, because the presence of the >C=O group can potentially modify the ring strain substantially. Accordingly, use of the values of $F_{ring}$ for cycloalkanes (Table 3) in conjunction with the parameters optimized above, results in calculated rate coefficients at 298 K that are generally overestimated for cycloketones. A set of adjusted values, denoted $F_{ring-CO}$, were therefore defined for 4-, 5- and 6-membered rings, based on the data for cyclobutanone, cyclopentanone and cyclohexanone (see Table 6). These are lower than those for cycloalkanes by respective factors of 5.1, 2.2 and 1.6, with a trend that suggests that the values of $F_{ring-CO}$ are once again tending towards unity as the size of the ring increases. The preferred data also include rate coefficients for four $C_9$ and $C_{10}$ terpenoids (camphenilone, camphor, nopinone and sabinaketone), which all contain bicyclic ketone structures. The corresponding rate coefficients calculated for these species using the optimized values of $F_{ring-CO}$ are in good agreement with the preferred data for camphenilone and camphor, but are underestimated by factors of 3.9 and 1.5 for nopinone and sabinaketone, respectively.



### 3.2.2 Hydroperoxides

The preferred data for the reactions of OH with saturated hydroperoxides are limited to recommended values for methyl hydroperoxide and $t$-butyl hydroperoxide, and a lower limit value for ethyl hydroperoxide. The temperature-dependent rate coefficient for H-atom abstraction from a hydroperoxy group, $k_{abs(-OOH)}$, was derived from the rate coefficient for the reaction of

OH with $t$-butyl hydroperoxide (Baasandorj et al., 2010), which provides the most direct measurement. The reported rate coefficient was corrected for (minor) reaction of OH at the methyl groups, with the assumption that there is no influence from the β-hydroperoxy group. The resultant Arrhenius parameters describing $k_{abs(-OOH)}$ are given in Table 4.

The limited data available suggest that a neighbouring hydroperoxy group has a significant activating effect on OH reactivity, as discussed previously (Jenkin et al., 1997; Saunders et al., 2003). In the present work, the value of F(-OOH) is assumed to be

identical to F(-OH), with the same value also assumed for peroxy linkages (denoted F(-OOR)), in the absence of kinetic and mechanistic data (see Table 5). Use of this value of F(-OOH) (in conjunction with the assigned value of $k_{abs(-OOH)}$) results in an underestimated rate coefficient for $CH_3OOH$, and a rate coefficient for $C_2H_5OOH$ that is at the recommended lower limit value (and therefore possibly also an underestimate). However, it overestimates the reported contribution of H-atom abstraction from the -CH(OOH)- group in the unsaturated isoprene-derived hydroperoxide, 2-hydroperoxy-3-methyl-but-3-en-1-ol, reported by St.

Clair et al. (2016), such that the assigned value of F(-OOH) appears to represent a reasonable compromise. The overall rate coefficients and calculated distributions of OH attack in $CH_3OOH$ and $C_2H_5OOH$ at 298 K are also in good agreement with the DFT calculations of Luo et al. (2011), which provides some additional support for the assigned parameters.

### 3.2.3 Ethers

The values of a number of parameters relevant to the oxidation of ethers are shown in Tables 7 and 8. These were optimized using

the preferred data for 14 acyclic mono-ethers, 13 acyclic di-ethers and 8 acyclic hydroxyethers. The original method of Kwok and Atkinson (1995) used the substituent factor F(-OR) to describe the effect of one or two α- ether linkages, with the influence of a β- ether linkage also subsequently considered in the review of Calvert et al. (2011). Both studies report difficulties in recreating the rate coefficients for the complete series of compounds, with discrepancies of up to over a factor of three between estimated and observed values. In the present work, the performance of the method is improved by defining a set of three rate coefficients

for H-atom abstraction from carbon atoms adjacent to ether linkages (see Table 7), which are applied independently of neighbouring group substituent factors. Similarly to Calvert et al. (2011), a substituent factor for β- ether groups, F(-CH2OR, -CH(OR)-, -C(OR)<), is also defined for application to H abstraction from other relevant sites in these compounds (see Table 8). These parameters were initially optimized for 298 K, and a correlation of $k_{calc}$ and $k_{obs}$ at 298 K is shown in Fig. 3. The updated method results in a value of $\Sigma((k_{calc}-k_{obs})/k_{obs})^2$ that is lower than that obtained by using the parameters reported by Calvert et al.

(2011), by a factor of 1.7. It reproduces the observed 298 K values for ethers, di-ethers and hydroxyethers to within factors of 1.4, 2.0 and 1.4, respectively.





Temperature-dependent recommendations are available for 22 of the above acyclic compounds. Of these, the data for 11 acyclic mono-ethers were used to provide optimized values of the temperature coefficients and pre-exponential factors for the group rate coefficients, $k_{abs(-OCH3)}$ and $k_{abs(-OR)}$ in Table 7 and the substituent factor F(-CH$_2$OR, -CH(OR)-, -C(OR)<) in Table 8. The data for 1,2-dimethoxyethane and 1,2-diethoxyethane were used to optimize the parameters for $k_{abs(-OCCOR)}$. The resultant $(E/R)_{calc}$ values are compared with the recommended $(E/R)_{obs}$ values in the lower panel of Fig. 3 (see also, Fig. S3).

The preferred data also include rate coefficients for seven cyclic mono-ethers and five cyclic di-ethers, which were not included in the optimization procedure described above. The limited dataset was used to define a set of $F_{ring-O}$ values for 3- to 7-membered rings containing one ether linkage, and a further set for 5-, 6- and 7-membered rings containing two ether linkages (see Table 6), with the values being applicable to 298 K. Temperature-dependent recommendations are available for three cyclic mono-ethers (5- to 7-membered rings) and four cyclic di-ethers (6- and 7-membered rings), which were used to optimize the corresponding values of $A_{F(ring-O)}$ and $B_{F(ring-O)}$ in Table 6.

### 3.2.4 Esters and carboxylic acids

Tables 7 and 8 also show the values of a number of parameters relevant to the oxidation of esters. These were optimized using the preferred data for 6 formates, 10 acetates, 12 higher esters, 5 dibasic esters, 2 hydroxy esters (lactates) and one carbonate. The original method of Kwok and Atkinson (1995) used the substituent factors F(-OC(=O)R) and F(-C(=O)OR) to describe the effects of ester groups, with a specific rate coefficient for H-atom abstraction from the formyl group in formate esters (denoted $k_{abs(ROCHO)}$ here) subsequently introduced by Le Calvé et al. (1997). In the present work, the method has been extended to include the parameter F(-CH$_2$C(=O)OR, -CH(C(=O)OR)-, -C(C(=O)OR)<) to represent the effect of a β-ester group, and the parameter F(-OC(=O)H) that is specific to formate esters. These parameters were initially optimized for 298 K, leading to the values given in Tables 7 and 8. A correlation of $k_{calc}$ and $k_{obs}$ at 298 K is shown in Fig. 4. The updated method reproduces the observed 298 K values for all the monobasic esters (formates, acetates and higher esters) and lactates to well within a factor of 2, although the rate coefficients for C$_4$-C$_7$ dibasic esters are generally overestimated (by factors in the range 2.2 – 4.0).

Temperature-dependent recommendations are available for 18 of the above compounds. In contrast to most of the preferred data, the preferred temperature dependences are described by a modified Arrhenius expression of the form, $k = A.T^2.\exp(-(E/R)/T)$. Optimization was achieved by a slightly modified procedure, in which values of both $k_{calc}$ and $k_{obs}$ were calculated at even 1/T intervals over the recommended temperature range for each ester (with an imposed upper limit of 400 K, where applicable), with the latter determined from the modified Arrhenius expression in each case. Representative values of $(E/R)_{calc}$ and $(E/R)_{obs}$ were then determined from a least squares linear regression of the data on a standard Arrhenius plot. The values of temperature coefficients of the relevant ester-specific parameters were varied to minimize the summed square deviation in the representative temperature coefficients, $\Sigma((E/R)_{calc}-(E/R)_{obs})^2$, leading to the values given in Tables 7 and 8. The resultant $(E/R)_{calc}$ values are compared with the recommended $(E/R)_{obs}$ values in the lower panel of Fig. 4 (see also, Fig. S4).

The preferred data also include rate coefficients for six carboxylic acids, which include all the C$_1$-C$_4$ alkanoic acids and pyruvic acid (2-oxo-propanoic acid). As shown in Table 8, the values of F(-C(=O)OR) and F(-CH$_2$C(=O)OR, -CH(C(=O)OR)-, -



C(C(=O)OR)<) optimized above are also assumed to apply to carboxylic acids (i.e. when -OR is -OH). The original method of Kwok and Atkinson (1995) defined a single rate coefficient for reaction at acid groups. In the present work, this is extended to a set of three rate coefficient, shown in Table 8. These include $k_{abs(formic\ acid)}$, which is specific to formic acid; and $k_{abs(RC(O)OH)}$, which is a generic parameter that applies to all higher acids, except those containing a 2-oxo group. In each case, the rate coefficient represents abstraction of the carboxyl H-atom. In the case of formic acid, abstraction of the formyl H-atom is represented by the parameter $k_{abs(ROCHO)}$ optimized above. For 2-oxo-carboxylic acids (e.g. pyruvic acid), a further rate coefficient, $k_{abs(RC(O)C(O)OH)}$, is defined for abstraction of the carboxyl H-atom, and the substituent group factor F(-C(=O)C(=O)OH) specific to this compound class is assigned a value of zero (see Table 8). The above parameters were initially optimized for 298 K, leading to the values given in Tables 7 and 8. A correlation of $k_{calc}$ and $k_{obs}$ at 298 K is included in Fig. 4.

Temperature-dependent recommendations are available for five of the above compounds. These were used to optimize the values of $E/R$ and $B_{F(X)}$ for the relevant parameters (Tables 7 and 8). The resultant $(E/R)_{calc}$ values are compared with the recommended $(E/R)_{obs}$ values in the lower panel of Fig. 4 (see also, Fig. S4).

### 3.2.5 Compounds containing oxidised nitrogen groups

The preferred data include rate coefficients for sets of compounds containing nitrate (or nitro-oxy) groups (-ONO$_2$) and nitro groups (-NO$_2$); and an upper limit value for peroxyacetyl nitrate (PAN, CH$_3$C(O)OONO$_2$). The first set contains data for 21 alkyl nitrates, 5 alkyl dinitrates, 12 hydroxyalkyl nitrates and 4 carbonyl nitrates, including both acyclic and cyclic compounds. The data for acyclic alkyl nitrates are the most extensive and well-determined, and these were used to optimize the 298 K values of the substituent factors F(-ONO$_2$) and F(-CH$_2$ONO$_2$, -CH(ONO$_2$)-, -C(ONO$_2$)<), leading to deactivating values are that are similar to those reported in the revised method of Atkinson (2000) (see Table 9). The correlation of $k_{calc}$ and $k_{obs}$ for acyclic alkyl nitrates (shown in Fig. 5) confirms that the trend in values is very well recreated. Fig 5 also compares $k_{calc}$ and $k_{obs}$ for a number of bifunctional nitrate classes, namely dinitrates, hydroxy-nitrates and carbonyl nitrates, and also cyclic compounds from all the considered classes. The values of $k_{calc}$ were determined using the above optimized substituent factors, and the relevant parameters optimized for other compound classes. The results indicate that the rate coefficients for acyclic dinitrates and hydroxynitrates are apparently systematically underestimated, whereas those for the cyclic compounds tend to be overestimated. Because the observed (preferred) values for these compounds are generally based on the results of single studies, the level of agreement is currently considered acceptable. Indeed, the reported rate coefficients for some acyclic dinitrates and hydroxynitrates would apparently require the nitrate group substituent factors to be activating, which is contrary to all published assessments. Further data on these compound classes would therefore be valuable.

Temperature-dependent recommendations are available for methyl nitrate, ethyl nitrate and 2-propyl nitrate. These data were used to provide optimized values of the temperature coefficients and pre-exponential factors for the nitrate group substituent factors, as shown in Table 9. The resultant $(E/R)_{calc}$ values are compared with the recommended $(E/R)_{obs}$ values in Fig. 5 (see also, Fig. S4).

The preferred data for compounds containing nitro groups include rate coefficients for a series of five nitroalkanes, based on the atmospheric pressure study of Nielsen et al. (1989). As discussed previously (e.g. Calvert et al., 2011), these rate coefficients are





systematically higher than those reported at low pressure (e.g. by Liu et al., 1990), particularly for nitromethane. This has been interpreted in terms of the reaction proceeding by partial addition of OH to the -NO$_2$ group (Kwok and Atkinson, 1995), with this represented by the rate coefficient $k_{add(-NO2)}$. The data were therefore used to optimize the 298 K values of the substituent factors F(-NO$_2$) and F(-CH$_2$NO$_2$, -CH(NO$_2$)-, -C(NO$_2$)<) given in Table 9, in conjunction with an optimized value of $k_{add(-NO2)} = 1.1 \times 10^{-13}$ cm$^3$ molecule$^{-1}$ s$^{-1}$. The resultant correlation of $k_{calc}$ and $k_{obs}$ for nitroalkanes is shown in Fig. 5. In the present work, the RN(OH)O$_2$ adduct formed from the addition component is assumed to decompose to yield HNO$_3$ and the organic radical, R. This is the only case where the reaction of OH with a saturated organic compound is not represented to result in abstraction of an H-atom from a C-H or O-H bond. For larger organic compounds containing nitro groups, however, this will generally account for a small fraction of the reaction (e.g. 4 % for 1-nitropentane).

The preferred data also include an upper limit rate coefficient for peroxyacetyl nitrate (PAN), based on the study of Talukdar et al. (1995). The value of the substituent factor F(-C(O)OONO$_2$) in Table 9 is set so that $k_{calc}$ is ≈ 50 % of the reported upper limit, which is consistent with the range of rate coefficients measured by Talukdar et al. (1995).

## 4 Unsaturated organic compounds containing C=C bonds

The reaction of OH with a given unsaturated organic compound can occur by both addition of OH to either side of each C=C bond, and by abstraction of H-atoms from the organic substituents. The estimated rate coefficient is therefore given by $k = k_{add} + k_{abs}$, where $k_{add}$ and $k_{abs}$ are summations of the partial rate coefficients for OH addition and H-atom abstraction for each attack position in the given organic compound.

The estimation of rate coefficients for H-atom abstraction ($k_{abs}$) makes use of the method and parameters optimized above for the reactions of OH with saturated organic compounds, with additional substituent factors defined to account for H-atom abstraction adjacent to C=C bonds, to form resonance-stabilized allyl-type radicals (as discussed further below). The estimation of rate coefficients for OH addition to C=C bonds ($k_{add}$) is based on the method described by Peeters et al. (2007), but is extended to include the effects of hydrocarbon and oxygenated substituent groups.

### 4.1 Alkenes and polyalkenes

### 4.1.1 Acyclic monoalkenes

For isolated C=C bonds in monoalkenes and polyalkenes, the Peeters et al. (2007) method defines site-specific parameters for addition to form primary, secondary and tertiary β-hydroxyalkyl radicals, as follows:

$$k(\text{-C=CH}_2) = k_{\text{prim-add}} \tag{8}$$

$$k(\text{-C=CH-X}) = k_{\text{sec-add}} \, F'(X) \tag{9}$$

$$k(\text{-C=C(-X)-Y}) = k_{\text{tert-add}} \, F'(X) \, F'(Y) \tag{10}$$



$k_{prim-add}$, $k_{sec-add}$ and $k_{tert-add}$ are the respective group rate coefficients for OH addition to form primary, secondary and tertiary β-hydroxyalkyl radicals; and F'(X) and F'(Y) are factors that account for the effects of the substituents X and Y. The reference substituent is defined as "-CH$_3$", such that F'(-CH$_3$) = 1.00. In the original work of Peeters et al. (2007), all alkyl and alkenyl substituents in monoalkenes and polyalkenes were also assigned a factor of F'(X) = 1.00, and this assumption is also largely applied in the present work. However, a small size-dependent substituent factor is considered for the specific case of acyclic linear alkyl substituents (-C$_n$H$_{2n+1}$), to help account for the reported increase of $k_{add}$ with alkene size for homologous series of alk-1-enes, 2-methyl-alk-1-enes and *trans*-alk-2-enes (Aschmann and Atkinson, 2008; Nishino et al., 2009).

The values of $k_{prim-add}$, $k_{sec-add}$ and $k_{tert-add}$ were initially optimized for 298 K, using the preferred kinetic data for the 44 acyclic monoalkenes in the preferred dataset. In general accordance with the analysis of Nishino et al. (2009), a value of F'(-C$_n$H$_{2n+1}$) = $(1 + \varepsilon[1-\exp(-0.35(C_n-1))])$ was applied for each linear alkyl substituent, where C$_n$ is the carbon number of the substituent and ε is a scaling factor. The relevant H-atom abstraction substituent factors, F(-C=CH$_2$), F(-C=CHR) and F(-C=CR$_2$), were also defined as part of the same procedure, where "R" denotes any alkyl group. The values of F(-C=CH$_2$) and F(-C=CHR) were constrained to obtain total branching ratios for H-atom abstraction of 6 % for but-1-ene and 3 % for *trans*-but-2-ene, as reported by Loison et al. (2010); and F(-C=CR$_2$) was assumed to be equal to F(-C=CHR). The values of $k_{prim-add}$, $k_{sec-add}$, $k_{tert-add}$ and ε were varied iteratively to minimize the summed square deviation, $\Sigma((k_{calc}-k_{obs})/k_{obs})^2$, for the set of alkenes. Fig. 6 shows a correlation of the optimized values of $k_{calc}$ with $k_{obs}$ at 298 K.

The resultant values of the optimized parameters are given in Tables 10, 11 and 12. The values of $k_{prim-add}$, $k_{sec-add}$ and $k_{tert-add}$ are slightly different from (but consistent with) those reported previously by Peeters et al. (2007), owing to optimization to the complete monoalkene dataset, and explicit consideration of H-atom abstraction. The optimized value of 0.14 for ε indicates that the enhancements in $k_{sec-add}$ and $k_{tert-add}$ are up to 14 % for each linear alkyl substituent. This effect is somewhat smaller than reported by Nishino et al. (2009), because the rate coefficients applied to account for H-atom abstraction from the alkyl groups in that study are smaller than those determined here. The values of F(-C=CH$_2$), F(-C=CHR) and F(-C=CR$_2$) in Table 12 are consistent with a significant activating influence on H atom abstraction adjacent to C=C bonds, resulting from the formation of resonance-stabilized radicals, as considered in detail previously in the DFT study of Vereecken and Peeters (2001). The corresponding rate coefficients for abstraction from primary, secondary and tertiary groups adjacent to a C=C bond thus lie in the respective ranges 0.11 – 0.27, 1.3 – 3.2 and 6.8 – 17 (in units of $10^{-12}$ cm$^3$ molecule$^{-1}$ s$^{-1}$ per H atom), which compare very well with the representative ranges calculated by Vereecken and Peeters (2001), 0.15 – 0.25, 1.5 – 3.0 and 8 – 15. Partly as a result of this, H-atom abstraction appears to be the dominant effect in accounting for the reported general increase in $k$ with alkene size for homologous series of alk-1-enes, 2-methyl-alk-1-enes and *trans*-alk-2-enes, as illustrated in Fig. S5. To a first approximation, therefore, the optimized size-dependent substituent factor for acyclic linear alkyl substituents (-C$_n$H$_{2n+1}$) can be considered as optional, and a value of F'(X) = 1.00 could alternatively be applied for simplicity.

Temperature-dependent recommendations are available for eight of the acyclic monoalkenes in Arrhenius format, as given in the preferred data in the Supplement (spreadsheet SI_3). These were used to provide optimized values of the temperature coefficient



($E/R$) and pre-exponential factor (A) for the group rate coefficients, $k_{prim-add}$, $k_{sec-add}$ and $k_{tert-add}$ (see Table 4), using the same procedure described above for the alkane H-atom abstraction reactions. The resultant $(E/R)_{calc}$ values are compared with the recommended $(E/R)_{obs}$ values in Fig. 6 (see also, Fig. S6).

### 4.1.2 Acyclic unconjugated (isolated) dienes

The parameter values determined above were also applied to calculate rate coefficients for the reactions of OH with six acyclic unconjugated (isolated) dienes (i.e. with remote C=C bonds) for which preferred kinetic data are available in the database. Three of these possess C=C bonds that are separated by a chain of two single C-C bonds, such that H-atom abstraction at the intermediate -CH$_2$- group forms a "superallyl" resonant structure. Because the two "-C=C-" substituents cannot therefore be regarded as independent, a relevant set of composite H-atom abstraction substituent factors, F((-C=CH$_2$)$_2$), F((-C=CH$_2$)(-

C=CHR)), F((-C=CH$_2$)(-C=CR$_2$)), F((-C=CHR)$_2$), F((-C=CHR)(-C=CR$_2$)) and F((-C=CR$_2$)$_2$) was defined, as indicated in Table 12. There are insufficient data to optimize these factors, and indeed only two of the six relevant structures are included in the set of unconjugated dienes. The factors were therefore assumed to be equal to the corresponding sum of those for formation of the component allyl structures, e.g. F((-C=CH$_2$)$_2$) = 2 × F(-C=CH$_2$), as shown in Table 12. As discussed further below (Sect 4.1.7), this assumption appears to provide reasonable estimates of branching ratios for H-atom abstraction, where

information is available. A correlation of the optimized values of $k_{calc}$ with $k_{obs}$ at 298 K is shown in Fig. 6. The optimized method reproduces all the observed values to within 23 %.

### 4.1.3 Cyclic alkenes and cyclic unconjugated dienes

The optimized parameter values were also used to estimate rate coefficients for the reactions of OH with 22 cyclic alkenes and cyclic unconjugated dienes for which preferred kinetic data are available in the database. For these calculations, no adjustments

were made for possible impacts of ring-strain or steric effects on the OH addition rate coefficients, although the empirical ring-strain factors, F$_{ring}$, determined above for 3-member through to 8-member rings were assumed to apply to the calculation of partial rate coefficients for H-atom abstraction. In addition to this, relevant tertiary (-CH<) groups at the bridgehead of strained bicyclic structures were assumed to be unable to form resonant allyl-type radicals upon abstraction of the H-atom, owing to the unfavourable orientation of the radical orbital, as discussed for α-pinene by Vereecken and Peeters (2001). In these specific cases,

the activating substituent factors in Table 12 were not applied.

A correlation of the optimized values of $k_{calc}$ with $k_{obs}$ at 298 K is shown in Fig. 6. The estimation method reproduces 15 of observed values to within 20 %, 18 to within 40 %, and all 22 values to within about a factor of two. In the four cases for which the absolute deviations are greater than 40 % (bicyclo[2.2.2]-oct-2-ene, α-pinene, sabinene and longifolene), it is not straightforward to rationalize the level of disagreement or modify the estimation method, because the deviations for some

structurally-similar compounds are either much smaller or in the opposite sense (e.g. α-pinene vs. 3-carene; sabinene vs. β-




pinene). Nevertheless, the level of performance of the estimation method can be regarded as acceptable, given that the series of compounds comprises complex bicyclic and polycyclic structures.

Temperature-dependent parameters are recommended for limonene, α-pinene and β-pinene in Arrhenius format. As shown in the lower panel of Fig. 6, the values of $E/R$ calculated from the parameters optimized above using the monoalkene dataset are in reasonable agreement with those observed (see also, Fig. S6).

### 4.1.4 Acyclic conjugated dienes

The estimation of rate coefficients for OH addition to conjugated diene systems is also based on the method described by Peeters et al. (2007). Site-specific rate coefficients for addition of OH to the internal carbon atoms of the diene system can be estimated using the parameters optimized above for monoalkenes. Addition of OH to the outer carbon atoms of the diene system generates resonance-stabilized hydroxy-substituted radicals, for which a further set of site-specific parameters is defined (see Peeters et al., 2007):

$$k(\text{-C=CH-C=CH}_2) = k_{\text{sec,prim}} \tag{11}$$

$$k(\text{-C=C(X)-C=CH}_2) = k_{\text{tert,prim}} \, (\text{F'(X)})^{\frac{1}{2}} \tag{12}$$

$$k(\text{-C=CH-C=CHX}) = k_{\text{sec,sec}} \, (\text{F'(X)})^{\frac{1}{2}} \tag{13}$$

$$k(\text{-C=C(X)-C=CHY}) = k_{\text{tert,sec}} \, (\text{F'(X) F'(Y)})^{\frac{1}{2}} \tag{14}$$

$$k(\text{-C=CH-C=C(X)Y}) = k_{\text{sec,tert}} \, (\text{F'(X) F'(Y)})^{\frac{1}{2}} \tag{15}$$

$$k(\text{-C=C(X)-C=C(Y)Z}) = k_{\text{tert,tert}} \, (\text{F'(X) F'(Y) F'(Z)})^{\frac{1}{2}} \tag{16}$$

The $k$ parameters (i.e. $k_{\text{sec,prim}}$ etc.) are the respective group rate coefficients for OH addition to form the corresponding resonance-stabilized radicals. In the first case, for example, the product is a resonance stabilized secondary ↔ primary radical:

HO-C-ĊH-C=CH$_2$ (secondary)  ↔  HO-C-CH=C-ĊH$_2$ (primary)

As above, F'(X), F'(Y) and F'(Z) are factors that account for the effects of the substituents X, Y and Z. Based on the limited data available for resonant radicals containing oxygenated substituents (presented in Sect. 4.2), the combined effect of the substituents is raised to the power of ½ for these resonant systems. This assumption has almost no effect for the dienes considered here because, with one exception, they contain no ≥ C$_2$ linear alkyl substituents such that F'(X) = 1.00 (in the exceptional case of *trans*-hexa-1,3-diene, there is a single ethyl group, which has a near-unity substituent factor, F'(X) = 1.04).

The values of the group rate coefficients were initially optimized for 298 K, using the preferred kinetic data for the 11 acyclic conjugated dienes in the database, with two of these (β-myrcene and β-ocimene) being trienes that possess an additional unconjugated C=C bond. Abstraction of an H-atom at a carbon atom adjacent to the conjugated diene system generates a





resonance-stabilized superallyl radical, and a corresponding set of H-atom abstraction substituent factors, F(-C=CH-C=CH$_2$), F(-C=C(R)-C=CH$_2$), F(-C=CH-C=CHR), F(-C=C(R)-C=CHR), F(-C=CH-C=CR$_2$) and F(-C=C(R)-C=CR$_2$) was therefore defined, as indicated in Table 12. Once again, the factors were assumed to be equal to the corresponding sum of those for formation of the component allyl structures, e.g. F(-C=CH-C=CH$_2$) = F(-C=CHR) + F(-C=CH$_2$), as shown in Table 12. The

corresponding rate coefficients for H-atom abstraction from primary, secondary and tertiary groups adjacent to a C=C-C=C bond system thus lie in the respective ranges 0.38 – 0.54, 4.5 – 6.4 and 24 – 34 (in units of $10^{-12}$ cm$^3$ molecule$^{-1}$ s$^{-1}$ per H atom), which compare reasonably well with the representative ranges calculated by Vereecken and Peeters (2001), 0.6 – 1.0, 6 – 10 and 30 – 60.

The value of one of the group rate coefficients ($k_{tert,tert}$) was left unchanged from that estimated by Peeters et al. (2007), owing to

the absence of kinetic data for acyclic conjugated dienes containing the relevant structure. The values of the other five ($k_{sec,prim}$, $k_{tert,prim}$, $k_{sec,sec}$, $k_{tert,sec}$ and $k_{sec,tert}$) were varied to minimize the summed square deviation, $\Sigma((k_{calc}-k_{obs})/k_{obs})^2$, for the set of acyclic conjugated dienes. The resultant values of the optimized parameters are given in Tables 10 and 12. The values of $k_{sec,prim}$, $k_{tert,prim}$, $k_{sec,sec}$ and $k_{sec,tert}$ are slightly different from (but consistent with) those reported previously by Peeters et al. (2007), owing to explicit consideration of H-atom abstraction in the present work. A correlation of the optimized values of $k_{calc}$ with $k_{obs}$ at 298 K

is shown in Fig. 6. The estimation method reproduces all the observed values to within 13 %.

Temperature-dependent recommendations are available for buta-1,3-diene and isoprene in Arrhenius format. These were used to optimize values of the temperature coefficient ($E/R$) and pre-exponential factor (A) for the group rate coefficients, $k_{sec,prim}$ and $k_{tert,prim}$ (see Table 10). The temperature dependences for both buta-1,3-diene and isoprene are well described by using a value of - 445 K for both $(E/R)_{sec,prim}$ and $(E/R)_{tert,prim}$, and this value was therefore also adopted for $(E/R)_{sec,sec}$, $(E/R)_{tert,sec}$, $(E/R)_{sec,tert}$ and

$(E/R)_{tert,tert}$. The values of the pre-exponential factors, A, were automatically returned from the corresponding $E/R$ and $k_{298K}$ values. The resultant $(E/R)_{calc}$ values are compared with the recommended $(E/R)_{obs}$ values in Fig. 6 (see also, Fig. S6).

### 4.1.5 Cyclic conjugated dienes

The optimized parameter values were also used to estimate rate coefficients for the reactions of OH with 5 cyclic conjugated dienes for which preferred kinetic data are available in the database. As above, no adjustments were made for the possible impacts of ring-strain or steric effects on the OH addition rate coefficients, but the empirical ring-strain factors, F$_{ring}$, determined

in Sect. 3.2 for 6- and 7-member rings were assumed to apply to the calculation of partial rate coefficients for H-atom abstraction. A correlation of the optimized values of $k_{calc}$ with $k_{obs}$ at 298 K is shown in Fig. 6. The estimation method reproduces the observed values for cyclohexa-1,3-diene, cyclohepta-1,3-diene and β-phellandrene to within 18 %. The deviations for the highly reactive monoterpenes α-phellandrene and α-terpinene are larger, the calculated values being about 30 % lower than those

observed.



### 4.1.6 Acyclic cumulative dienes

Preferred kinetic data are available for the reactions of OH with four cumulative dienes, namely propadiene, buta-1,2-diene, penta-1,2-diene and 3-methyl-buta-1,2-diene. Addition of OH to these structures cannot be described by the parameters defined above, so a further set of site-specific parameters is defined here, as summarized in Table 1. $k_v$ is a generic group rate coefficient describing OH addition to each of the outer carbons atom of the diene system, leading to the formation of an alkenyl (vinyl) radical. Because the substitution of the radical site is invariant, this rate coefficient is assumed to be identical in all cases. The other $k$ parameters (i.e. $k_{pp}$ etc.) are the respective group rate coefficients for OH addition to the central carbon atom, which leads to the radical centre being on either of the two outer carbon atoms. The subscripts (p = primary; s = secondary; t = tertiary) describe the level of substitution of the possible product radical centres. The parameter $k_{pp}$ is specific to propadiene, with the total rate coefficient being $k_{pp} + 2k_v$. Daranlot et al. (2012) have inferred that the reaction occurs 80 % via addition to the internal carbon atom, based on a combination of experimental results and theoretical calculations. This branching ratio was therefore used to constrain the relative values of $k_{pp}$ and $k_v$ in the present work, i.e. $k_{pp} = 8k_v$.

The values of the group rate coefficients were optimized for 298 K, using the preferred kinetic data for the four cumulative dienes. Abstraction of an H-atom at a carbon atom adjacent to the diene system potentially generates a resonant radical. However, because of the vinyl character of one of the resonant forms, the corresponding substituent factor, F(-C=C=C<), is assumed not to be activating (see Table 12). As above, the appropriate value of F'(R) was applied to account for the activating effect of linear - $C_nH_{2n+1}$ groups, although this only results in a very small adjustment in the one case of penta-1,2-diene, and is therefore not fully tested by the current dataset.

The values of $k_v$, $k_{pp}$, $k_{ps}$ and $k_{pt}$ were varied to minimize the summed square deviation, $\Sigma((k_{calc}-k_{obs})/k_{obs})^2$, for the set of cumulative dienes. The resultant values of the optimized parameters are given in Table 10. A correlation of the optimized values of $k_{calc}$ with $k_{obs}$ at 298 K is shown for the four cumulative dienes in Fig. 6. The estimation method reproduces the observed values to within 13 %. The values of the other parameters ($k_{ss}$, $k_{st}$ and $k_{tt}$) could not be optimized, owing to the absence of the relevant structures in the set of compounds for which data are available. However, it is noted that $k_{pp} \approx (2k_{prim-add})$, $k_{ps} \approx (k_{prim-add} + k_{sec-add})$ and $k_{pt} \approx (k_{prim-add} + k_{tert-add})$. The values of $k_{ss}$, $k_{st}$ and $k_{tt}$ are therefore provisionally set to be approximately the sum of the corresponding combinations of $k_{sec-add}$ and $k_{tert-add}$. Data for larger cumulative dienes are required to test this assumption.

A temperature-dependent recommendation is available for propadiene in Arrhenius format. A corresponding rounded value of $E/R$ was therefore assigned to both $(E/R)_v$ and $(E/R)_{pp}$ (see Table 10). In all the other cases, the provisional $E/R$ values are based on the weighted average of those for the corresponding combinations of $k_{prim-add}$, $k_{sec-add}$ and $k_{tert-add}$. The values of the pre-exponential factors, A, were automatically returned from the corresponding $E/R$ and $k_{298K}$ values.

As indicated above, addition of OH to the central carbon atom of a cumulative diene system leads to the radical centre being on either of the two outer carbon atoms. In the absence of data, the formation ratio of the two possible radical products in asymmetric systems is also based on the relative values of the relevant rate coefficients, $k_{prim-add}$, $k_{sec-add}$ and $k_{tert-add}$, leading to the more substituted product radical being favoured. Clearly additional information is required to confirm this approach.



### 4.1.7 Branching ratios for H-atom abstraction

The site-specific partial rate coefficients estimated by the above methods also define the branching ratios for both OH addition and H-atom abstraction for the reaction of OH with a given alkene. The total 298 K branching ratios for H-atom abstraction, $k_{abs}/(k_{abs} + k_{add})$, are presented for all compounds in Fig. S7, which are calculated to lie in the range $0 - 33$ % using the methods

presented above. These values suggest that, although OH addition remains the dominant process for all the compounds, H-atom abstraction is potentially significant in many cases. Reported branching ratios for H-atom abstraction are available for a subset of nine of the compounds considered in the present work. The values are listed in Table 13, along with the corresponding 298 K values calculated by the SAR method presented here. The values are also compared in a correlation plot, shown in Fig. S8, confirming that the SAR broadly recreates the trend in the observed values.

## 4.2 Organic oxygenates containing C=C bonds

The preferred 298 K data include rate coefficients for reactions of OH with 81 unsaturated oxygenated compounds containing C=C bonds. These include data for 18 alcohols, 16 aldehydes, 17 ketones and hydroxyketones, 2 hydroxy-hydroperoxides, 13 esters, 1 acid (propenoic acid), 7 hydroxy-nitrates, 2 dinitrates, 1 carbonyl nitrate (*trans*-2-methyl-4-nitrooxy-2-buten-1-al), 1 peroxyacyl nitrate (MPAN) and 3 nitroalkenes. In practice, only five of these compounds contain conjugated double bonds, with

15 the oxygenated substituents limited to aldehyde and ketone groups. As a result, the methods optimized below are mainly based on the impacts of oxygenated groups on isolated double bonds and, in some cases, are derived from very sparse datasets.

Table 11 presents substituent factors, F'(X), for a variety of oxygenated substituents, which each quantify the effect of replacing a -CH$_3$ substituent with the given group. These were initially optimized for 298 K, by minimizing the summed square deviation, $\Sigma((k_{calc}-k_{obs})/k_{obs})^2$, for the sets of compounds summarized in the notes to Table 11. This procedure also took account of the

20 contributions from H atom abstraction reactions at relevant sites within the compounds, using the methods presented above. For H-atom abstraction adjacent to C=C bonds (forming resonance-stabilized allyl-type radicals), the factors for alkenes and dienes (e.g. F(-C=CHR)) given in Table 12 were modified to account for the effects of oxygenated substituents to the double bonds, using the relevant values of the H-atom abstraction substituent factors, F(X), given in Tables 5, 8 and 9, for example:

$$F(-C=CHX) = F(-C=CHR) \, F(X) \tag{17}$$

$$F(-C=C(X)-C=CHY) = F(-C=C(R)-C=CHR) \, (F(X) \, F(Y))^{\frac{1}{2}} \tag{18}$$

Parameters calculated in this way currently only apply to a limited number of unsaturated oxygenates for which kinetic data are available, and the corresponding abstraction routes generally make relatively minor contributions to the overall calculated rate coefficient. As a result, this approach must be regarded as provisional, with further information required for its full validation. In the specific case of H-atom abstraction from a formyl group adjacent to a C=C bond, formation of a resonant radical is not

represented, and a single rate coefficient ($k_{abs(-CHO)-\alpha C=C}$) was simultaneously optimized, based on data for 13 $\alpha,\beta$-unsaturated



aldehydes (see Table 4). As with the other formyl group rate coefficients in Table 4, this rate coefficient is applied independently of substituent factors. A correlation of the resultant values of $k_{calc}$ with $k_{obs}$ at 298 K is shown in the upper panel of Fig. 7. Temperature-dependent recommendations are available for a subset of 22 unsaturated organic oxygenates. Where possible, these were used to provide representative values of the temperature coefficients ($B_{F(X)}$) and pre-exponential factors ($A_{F(X)}$) for the

substituent factors given in Table 2. The values of $B_{F(X)}$ were varied with the aim of minimizing the summed square deviation in the composite temperature coefficients, $\Sigma((E/R)_{calc}-(E/R)_{obs})^2$, for the contributing sets of compounds. The resultant $(E/R)_{calc}$ values are compared with the recommended $(E/R)_{obs}$ values in the lower panel of Fig. 7 (see also Fig. S9). The values of $A_{F(X)}$ were automatically returned from the corresponding optimized $B_{F(X)}$ and $F(X)_{298K}$ values. An optimized temperature-dependence expression for $k_{abs(-CHO)-\alpha C=C}$ was also determined as part of this procedure, as given in Table 4.

The site-specific partial rate coefficients estimated by the above SAR methods can also be used to define the branching ratios for both OH addition and H-atom abstraction for the reaction of OH with a given unsaturated oxygenate. Where available, the present methods appear to provide a reasonable representation of reported product yields and mechanistic information (e.g. see examples given in the Supplement).

## 5 Unsaturated organic compounds containing C≡C bonds

Reported kinetic data for the reactions of OH with alkynes are available for ethyne (acetylene), propyne, but-1-yne, but-2-yne, pent-1-yne and hex-1-yne. These data suggest that the rate coefficient for OH addition to C≡C bonds in alkynes cannot be estimated in an analogous way to that applied to alkenes above (i.e. by adding partial rate coefficients for addition of OH to each side of the triple bond). In this case, the addition rate coefficient is based on a single parameter for the C≡C group ($k_{C≡C}$), which is modified on the basis of the identities of the two substituent groups:

$k(\text{X-C≡C-Y}) = k_{C≡C} \, F_{C≡C}(X) \, F_{C≡C}(Y)$                              (19)

The values of $k_{C≡C}$ and relevant $F_{C≡C}(X)$ were optimized using the preferred dataset, with rate coefficients based on high pressure limiting values. For this procedure, $F_{C≡C}(-H)$ was assigned a value of 1.00, and the abstraction of H atoms from the substituent alkyl groups was treated using the method optimized above in Sect. 3, with $F(-C≡C-)$ also assumed to take a value of 1.00 (as previously applied by Kwok and Atkinson, 1995). This resulted in an optimized value of $k_{C≡C} = 9.4 \times 10^{-13}$ cm$^3$ molecule$^{-1}$ s$^{-1}$,

with $F_{C≡C}(-CH_3) = 4.8$ for a methyl substituent, and $F_{C≡C}(-R) = 8.0$ applied to all other alkyl substituents (although the data are limited to alkynes possessing $C_1$-$C_4$ linear alkyl substituents). Fig. 8 shows a correlation of the optimized values of $k_{calc}$ and $k_{obs}$, demonstrating that the trend of rate coefficients for this series of alkynes is well reproduced using these parameters. The kinetics have been reported to be only weakly dependent on temperature at high pressures (Boodaghians et al., 1987; Zádor and Miller, 2015) and the above values of $k_{C≡C}$, $F_{C≡C}(-CH_3)$ and $F_{C≡C}(-R)$ are therefore assumed to apply over the tropospheric temperature

range.



The addition of OH can potentially occur at the carbon atoms on either side of the triple bond. Product yields reported for propyne in some experimental studies (Hatakeyama et al., 1986; Lockhart et al., 2013), and a theoretical appraisal of the propyne system (Zádor and Miller, 2015), suggest that formation of the more substituted product radical is strongly favoured, but with evidence for addition to both sides of the C≡C bond reported by Yeung et al. (2005). It is therefore assumed that the ratios

for formation of the product radicals, HO-C(-Y)=Ċ(-X) and HO-C(-X)=Ċ(-Y), are given by $F_{C≡C}(X)/(F_{C≡C}(X) + F_{C≡C}(Y))$ and $F_{C≡C}(Y)/(F_{C≡C}(X) + F_{C≡C}(Y))$, respectively. However, this provisional assumption is based on limited information, and further products studies for the reactions of OH with asymmetric alkynes are required to test this approach.

At present, data for compounds containing both a C≡C bond and an oxygenated substituent appear to be limited to prop-2-yn-1-ol and 3,5-dimethyl-hex-1-yn-3-ol. Based on the 298 K preferred values of $k_{obs}$ for these compounds, the presence of a hydroxy

substituent on the carbon atom adjacent to the C≡C bond has an additional optimized activating effect of a factor of 3.5 relative to the values of $F_{C≡C}(-CH_3)$ and $F_{C≡C}(-R)$ indicated above. The resultant values of $k_{calc}$, based on this enhancement, are compared with $k_{obs}$ in Fig. 8. Clearly additional data are also required to confirm the reliability of this provisional estimate, and to allow factors for a variety of oxygenated substituents to be defined.

## 6 Reactions of organic radicals with $O_2$ and competing processes

Carbon-centred organic radicals (R) formed from the reactions that initiate VOC degradation (or from other routes, such as decomposition of larger oxy radicals) can react with molecular oxygen ($O_2$) under tropospheric conditions, to form the corresponding thermalized peroxy radicals ($RO_2$), the chemistry of which will be summarized elsewhere (Jenkin et al., 2018b):

R + $O_2$ (+M) → $RO_2$ (+M)                                                                                    (R1)

(M denotes a third body, most commonly $N_2$ or $O_2$ under atmospheric conditions). Rate coefficients for organic radicals containing three or more heavy atoms (i.e. C, O and N) are expected to be close to the high-pressure limit under tropospheric conditions. Table 14 (comment (a)) shows representative values of the rate coefficients (based on $C_4$ alkyl radicals), which are consistent with reaction (R1) typically occurring on a timescale of ≤ 25 ns in air at atmospheric pressure. In the absence of competing processes, reaction (R1) therefore does not need to be included explicitly in atmospheric mechanisms, which is

the case for the large majority of R. The remainder of this section summarizes the exceptions to this rule, where either R, or the initially-formed peroxy radical adduct, $[ROO]^‡$, undergoes competitive or exclusive decomposition or rearrangement. In addition, the treatment of reaction (R1) for systems with an allyl resonance (i.e. where $O_2$ can add at two positions) is also described. Abstraction of a hydrogen atom from hydroxy and hydroperoxy groups in VOCs results in formation of organic oxy and peroxy radicals, respectively. The treatment of these species will be summarized elsewhere (e.g. Jenkin et al.,

2018b).





## 6.1 Competitive decomposition or rearrangement of R

Table 14 summarizes the instances where the thermalized organic radical, R, is represented to undergo a prompt decomposition or rearrangement that is either its exclusive fate under atmospheric conditions, or is competitive with reaction (R1). Organic radicals with -OOH, -OOR' (where R' is an organic group) or -ONO$_2$ groups α- to the radical centre are

5 estimated to decompose spontaneously on the picosecond timescale (Vereecken et al., 2004; Vereecken, 2008), as shown in Table 14. These processes can therefore be assumed to occur exclusively for all relevant organic radicals. The other processes shown in Table 14 are estimated to compete with addition of O$_2$ (reaction (R1)), and the rate coefficient ratios allow the relative importance of the two processes to be represented in each case.

In some cases, organic radicals formed specifically from the reactions of OH with VOCs are formed chemically activated,

[R]$^{\ddagger}$, and the rate of decomposition or rearrangement is enhanced. These are represented as follows:

(i) Abstraction of the formyl H atom in methylglyoxal (CH$_3$C(=O)CHO) via reaction with OH has been reported to generate activated [CH$_3$C(=O)CO]$^{\ddagger}$ radicals which decompose promptly and exclusively (Baeza-Romero et al., 2007). This is therefore also assumed for [R'C(=O)CO]$^{\ddagger}$ formed specifically from the reactions of OH with higher analogues (where R' is any organic group), leading to the following overall reaction:

OH + R'C(=O)CHO  → R' + CO + CO (+ H$_2$O) (40 %)  (R2a)

→ R'Ċ=O + CO (+ H$_2$O)  (60 %)  (R2b)

For thermalized R'C(=O)ĊO radicals, formed via other routes (e.g. decomposition of larger oxy radicals), decomposition is assumed to occur in competition with reaction with O$_2$, as shown in Table 14, based primarily on the results of Jagiella and Zabel (2008) for thermalized CH$_3$C(=O)Ċ=O radicals.

(ii) The addition of OH to unsaturated VOCs generates chemically activated β-hydroxy organic radicals. In most cases, these subsequently become fully thermalized under atmospheric conditions, and react exclusively with O$_2$ via reaction (R1) to form the corresponding β-hydroxy peroxy radicals. In a few cases, however, prompt rearrangements are represented to compete with stabilization, as shown in Table S1. These specifically include structures where the radical centre is on the carbon atom adjacent to a cyclopropyl, oxiranyl or gem-disubstituted cyclobutyl ring, radicals formed from the addition of

OH to the central carbon atoms of conjugated dienes, and structures where the radical centre is on the carbon atom adjacent to an -OOH, -C(=O)OOH or -C(=O)OONO$_2$ group.

## 6.2 Competitive decomposition or rearrangement of chemically activated [ROO]$^{\ddagger}$ adducts

Table 15 summarizes the instances where chemically activated [ROO]$^{\ddagger}$ adducts, formed initially from the reactions of specific organic radicals with O$_2$, are represented to undergo a prompt decomposition or rearrangement that is either its exclusive fate

under atmospheric conditions, or competes with stabilization to form the thermalized peroxy radical, RO$_2$. These specifically



include those formed from the reactions of $O_2$ with α-hydroxy organic radicals, vinyl radicals, 2-hydroxyvinyl radicals and cyclohexadienyl radicals.

(i) The reactions of $O_2$ with α-hydroxy organic radicals, $>\dot{C}OH$, are reported to form both chemically activated $[>C(OH)OO]^{\ddagger}$ adducts, and thermalized peroxy radicals, $>C(OH)O_2$, with the yield of the latter increasing with radical size. The chemically

activated $[>C(OH)OO]^{\ddagger}$ adducts are estimated to isomerize and decompose promptly and exclusively (i.e. on the sub-nanosecond timescale) as follows (Dibble, 2002; Capouet et al., 2004; Hermans et al., 2005):

$$>\dot{C}OH + O_2 \rightarrow [>C(OH)OO]^{\ddagger} \rightarrow >C(=O) + HO_2 \qquad\qquad (R3)$$

As will be discussed in more detail elsewhere (Jenkin et al., 2018b), the thermalized $>C(OH)O_2$ radicals can also isomerize and decompose to form a carbonyl product (denoted $>C(=O)$) and $HO_2$, and this may also be the dominant fate under atmospheric

conditions in many cases. However, this occurs on millisecond timescales, such that other competitive isomerization reactions may need to be considered for specific peroxy radical structures, and bimolecular reactions (e.g. with NO) can compete for all such peroxy radicals under chamber conditions with ppm levels of $NO_x$. Evidence for the formation of thermalized peroxy radicals has been reported in both laboratory studies (e.g. Orlando et al., 2000; Jenkin et al., 2005; Aschmann et al., 2010) and theoretical studies (Capouet et al., 2004; Hermans et al., 2005), with the data suggesting that the fraction of thermalized radicals

increases with radical size (see Table 16 and Fig. S10). Based on this information, the fraction of thermalized radicals (β) is provisionally defined in terms of the number of heavy (C, O and N) atoms the organic group (R) contains, denoted $n_{CON}$, as follows: β = 0 for $n_{CON} \leq 5$; β = $[1+\exp(-0.75(n_{CON}-10))]^{-1}$ for $6 \leq n_{CON} \leq 14$; and β = 1 for $n_{CON} \geq 15$. It is noted that this representation is based on a limited dataset, and that further systematic information is required to refine the structural dependence of fractional formation of thermalized α-hydroxy peroxy radicals.

(ii) The reactions of $O_2$ with vinyl radicals, $>C=\dot{C}R'$, form chemically activated $[>C=C(R')OO]^{\ddagger}$ adducts, which isomerize and decompose to form a carbonyl product and a chemically activated acyl radical (e.g. Carpenter, 1995; Eskola and Timonen, 2003; Matsugi and Miyoshi, 2014):

$$>C=\dot{C}R' + O_2 \rightarrow [>C=C(R')OO]^{\ddagger} \rightarrow >C(=O) + [R'\dot{C}=O]^{\ddagger} \qquad\qquad (R4)$$

The chemically activated acyl radical, $[R'\dot{C}=O]^{\ddagger}$, is represented to either decompose to form R' and CO (65 %) or to be stabilized

to form $R'\dot{C}O$ (35 %), leading to the overall chemistry shown in Table 15. These ratios are based observations for the reaction of $O_2$ with the methylvinyl radical, formed during the OH-initiated oxidation of methacrolein (Orlando et al., 1999), although dominant decomposition of $[H\dot{C}=O]^{\ddagger}$, formed from the reaction of $O_2$ with the vinyl radical, has also been reported (Matsugi and Miyoshi, 2014). In the absence of additional systematic data, these product ratios are applied generally to the reactions of $O_2$ with vinyl radicals, with the exception of 2-hydroxyvinyl radicals, which are considered below.

(iii) The reactions of $O_2$ with 2-hydroxyvinyl radicals, $-C(OH)=\dot{C}R'$ (formed, for example, from the addition of OH to alkynes), form chemically activated $[-C(OH)=C(R')OO]^{\ddagger}$ adducts. Based on the products reported for the OH initiated oxidation of several



alkynes (e.g. Hatakeyama et al., 1986; Yeung et al., 2005; Lockhart et al., 2013), $[-C(OH)=C(R')OO]^{\ddagger}$ is represented to isomerize and decompose via two pathways as follows (leading to the overall chemistry shown in Table 15):

$$[-C(OH)=C(R')OO]^{\ddagger} \quad\quad \rightarrow -C(=O)C(=O)R' + OH \quad (70\ \%) \quad\quad\quad\quad (R5a)$$

$$\rightarrow -C(=O)OH + R' + CO \quad (30\ \%) \quad\quad\quad\quad (R5b)$$

The assigned product ratios are based primarily on the OH yields reported by Lockhart et al. (2013) for ethyne, propyne and but-2-yne, but are also informed by the observations of α-dicarbonyls ($-C(=O)C(=O)R'$) and carboxylic acids ($-C(=O)OH$) reported by Hatakeyama et al. (1986) and Yeung et al. (2005).

(iv) The reactions of $O_2$ with cyclohexadienyl and alkyl-substituted cyclohexadienyl radicals (formed from the abstraction of an H atom from cyclohexadiene and alkyl-substituted cyclohexadienes), have been reported to generate an aromatic hydrocarbon

product and $HO_2$ in a number of studies (Ohta et al., 1984; Tuazon et al., 2003; Jenkin et al., 2005; Aschmann et al., 2011), with the reaction proceeding either via formation of an $[ROO]^{\ddagger}$ adduct, or via a direct H atom abstraction mechanism. Based on those studies, this reaction channel is represented to occur exclusively for this radical class. As discussed in the companion paper (Jenkin et al., 2018a), the same process also partially occurs for hydroxy-substituted cyclohexadienyl radicals formed from the addition of OH to aromatics, but with other pathways also contributing in those cases.

**6.3 Reversible addition of $O_2$ to allyl radicals**

If an organic radical possesses an allyl resonance, there are two possible addition sites for $O_2$. Furthermore, the reverse decomposition of the two $RO_2$ radicals to reform the allyl radical is reported to occur at a rate that is competitive with those for the alternative reactions that are available to the $RO_2$ radicals under typical atmospheric conditions. This therefore needs to be taken into account when representing the reactions of $O_2$ with asymmetric allyl radicals, because the relative formation of the two

$RO_2$ radicals may depend on the prevailing atmospheric conditions.

The reversible addition of $O_2$ to allyl radicals can be represented schematically as follows (substituents have been omitted for clarity):

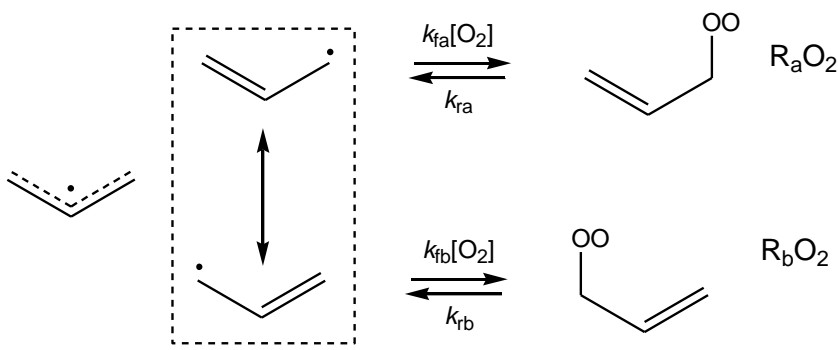



The total rate coefficient for addition of $O_2$ is given by $(k_{fa} + k_{fb})$, where the terms represent partial rate coefficients for the association reactions forming $R_aO_2$ and $R_bO_2$, respectively. The reverse rate coefficients, $k_{ra}$ and $k_{rb}$, characterise the decomposition rates of the individual peroxy radical structures.

Reported experimental kinetic and thermodynamic data are limited to information on the reactions of $O_2$ with the two simplest allyl radicals, $CH_2CHCH_2$ (allyl) and $CH_3CHCHCH_2$ (1-methylallyl) (Ruiz et al., 1981; Morgan et al., 1982; Jenkin et al., 1993; Knyazev and Slagle, 1998; Rissanen et al., 2012). This information allows representative rate coefficients to be defined for forward and reverse reactions for alkyl-substituted allyl radical + $O_2$ systems, as summarized in Tables 17 and 18.

Peeters et al. (2014) have estimated parameters for a set of hydroxy-substituted allyl radicals formed from the addition of OH to isoprene, using a combination of DFT and ab initio methods. Suggestions for refinements were subsequently made by Peeters (2015), taking account of provisional laboratory results reported by Crounse et al. (2014). Those recommendations (given in Table S2) were previously adopted for use in MCM v3.3.1 (Jenkin et al., 2015), and remain the preferred values for the hydroxyalkyl-substituted allyl and allyl peroxy radicals formed specifically from the addition of OH to isoprene. Because the addition of OH to conjugated dienes represents an important source of allyl radicals, the information has also been used to define approximate rate coefficients for a generic set of hydroxyalkyl-substituted allyl and allyl peroxy radicals for provisional application to other systems, which are also summarized in Tables 17 and 18.

The treatment of allyl radicals containing a number of oxygenated substituents is significantly simplified. Addition of $O_2$ is assumed to occur exclusively (and irreversibly) at the site possessing the substituent that is higher in the following list: -OH/-OR/-OOH/-OOR > -OC(=O)H/-OC(=O)R > alkyl/-H > -C(=O)H/-C(=O)R > -C(=O)OH/-C(=O)OR > -ONO$_2$ > -NO$_2$. If both sites possess an oxygenated substituent of the same rating, $O_2$ addition is assumed to occur equally at each site. For other allyl radicals containing substituents with more remote oxygenated groups, the rate coefficients for alkyl-substituted allyl radical + $O_2$ systems given in Tables 17 and 18 are used as a default.

## 7 Conclusions

Updated and extended structure activity relation (SAR) methods have been developed to estimate rate coefficients for the reactions of the OH radical with aliphatic organic species. The group contribution methods were optimized using a database including a set preferred rate coefficients for 489 species. The overall performance of the SARs in determining $\log k_{298K}$ is now summarized.

The distribution of errors ($\log k_{calc}/k_{obs}$), the Root Mean Squared Error (RMSE), the Mean Absolute Error (MAE) and the Mean Bias Error (MBE) were examined to assess the overall reliability of the SAR. The RMSE, MAE and MBE are here defined as:

$$RMSE = \sqrt{\frac{1}{n}\sum_{i=1}^{n}(\log k_{calc} - \log k_{obs})^2} \qquad (20)$$



$$MAE = \frac{1}{n}\sum_{i=1}^{n}|\log k_{calc} - \log k_{obs}| \quad\quad\quad (21)$$

$$MBE = \frac{1}{n}\sum_{i=1}^{n}(\log k_{calc} - \log k_{obs}) \quad\quad\quad (22)$$

where $n$ is the number of species in the dataset. The assessment was performed for various subsets to identify possible biases within a category of species (e.g. saturated vs unsaturated, cyclic vs acyclic, hydrocarbons vs functionalized species). Errors computed for the various subsets are summarized Fig. 9 for hydrocarbons, Fig. 10 for monofunctional species and Fig. 11 for the full set of species.

The calculated $\log k_{298K}$ shows no significant bias, with MBE remaining below 0.05 log units for the various subsets, and with median values of the error distributions close to zero (see Figs. 9-11). For the hydrocarbons, the SARs show similar performances for the alkane and the alkene subsets, with a RMSE of the order of 0.05 and 0.10 log units for acyclic and cyclic species, respectively (see Fig. 9). For monofunctional species, RMSE ranges from 0.07 (aldehyde subset) to 0.21 (nitro subset) (see Fig. 10). For this category of species, the SAR provides better estimates for the saturated subset of species (RMSE = 0.11) compared to the unsaturated subset (RMSE = 0.16). For the full database, however, the SARs show similar performances for both cyclic/acyclic structures and saturated/unsaturated carbon skeletons (see Fig. 11). Fig. 11 also shows that the reliability of the SARs decreases with the number of functional groups on the carbon skeleton. Indeed, the RMSE increases from 0.07 for hydrocarbons to 0.13 for monofunctional species and reaches 0.22 for multifunctional species, i.e. a relative error for the calculated $k_{298K}$ of 17 %, 35 % and 66 %, respectively. In the multifunctional subset (124 species), most of the species are bifunctional compounds (116 species), with a limited contribution from trifunctional compounds. The reliability of the SARs for species with more than two functional groups can therefore not be assessed. The atmospheric oxidation of hydrocarbons and organic oxygenates likely leads to a myriad of highly functionalized species (e.g. Aumont et al., 2005, 2012, Goldstein and Galbally, 2007; Mentel et al., 2015). Extrapolation of the SAR to this category of compound is therefore required in models aiming to describe atmospheric oxidation explicitly. Additional rate coefficients would therefore be highly valuable for further assessment and constraining of SARs for multifunctional species. Finally, for the full database, the SARs give fairly reliable $k_{298K}$ estimates, with a MAE of 0.09 and a RMSE of 0.15, corresponding to an overall agreement of the calculated $k_{298K}$ within 40%.

## Acknowledgements

This work received funding from the Alliance of Automobile Manufacturers, and as part of the MAGNIFY project, with funding from the French National Research Agency (ANR) under project ANR-14-CE01-0010, and the UK Natural Environment Research Council (NERC) via grant NE/M013448/1. Marie Camredon (LISA, Paris) is gratefully acknowledged for helpful discussions on this work.



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





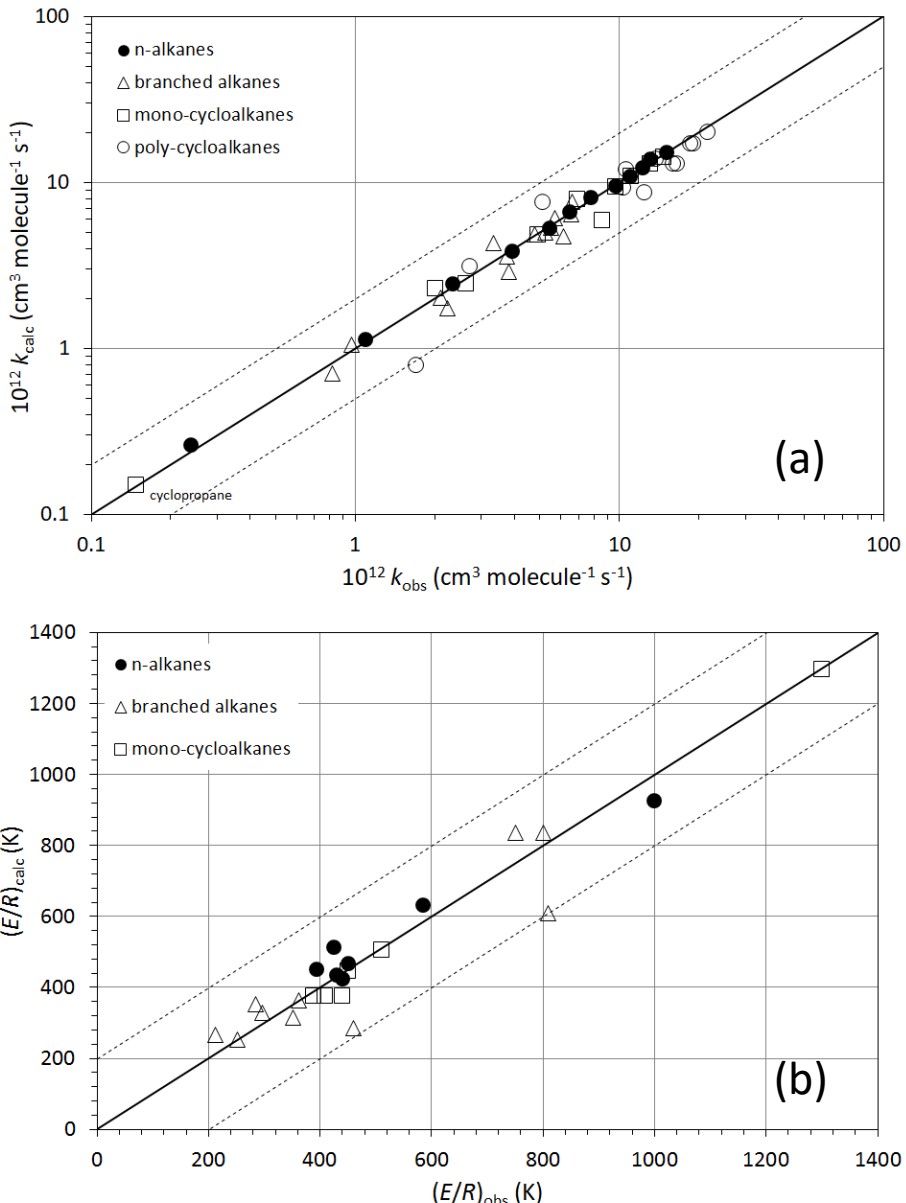

**Figure 1: (a) A log-log correlation of $k_{calc}$ and $k_{obs}$ at 298 K for alkanes (for presentation purposes, the value for cyclopropane has been scaled up by a factor of two). The broken lines show the ± a factor of 2 range; (b) A correlation of the temperature coefficients $(E/R)_{calc}$ and $(E/R)_{obs}$ for alkanes. The broken lines show the ± 200 K range.**





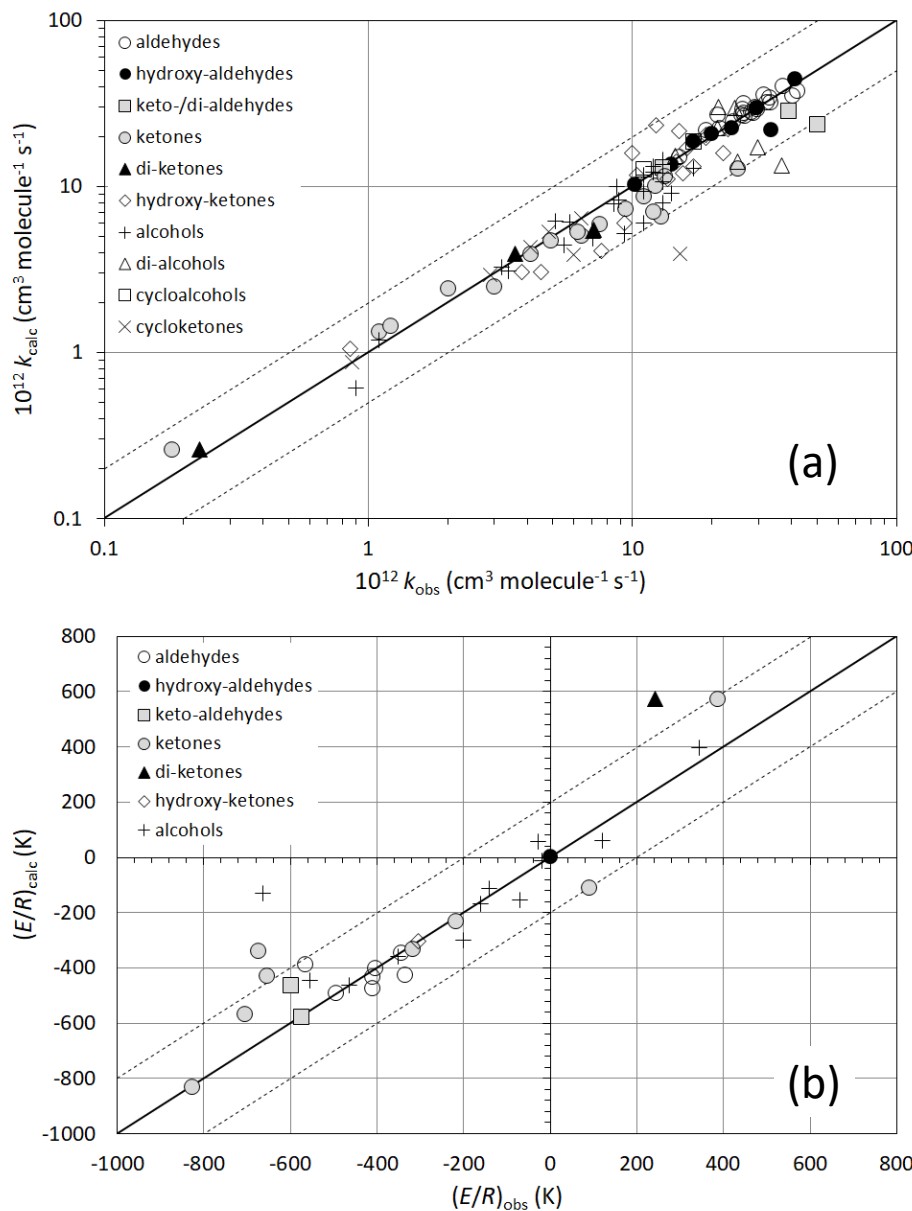

**Figure 2: (a) A log-log correlation of $k_{calc}$ and $k_{obs}$ at 298 K for saturated organic oxygenates containing carbonyl and hydroxy groups. The broken lines show the ± a factor of 2 range; (b) A correlation of the temperature coefficients $(E/R)_{calc}$ and $(E/R)_{obs}$ for the same compound classes. The broken lines show the ± 200 K range.**



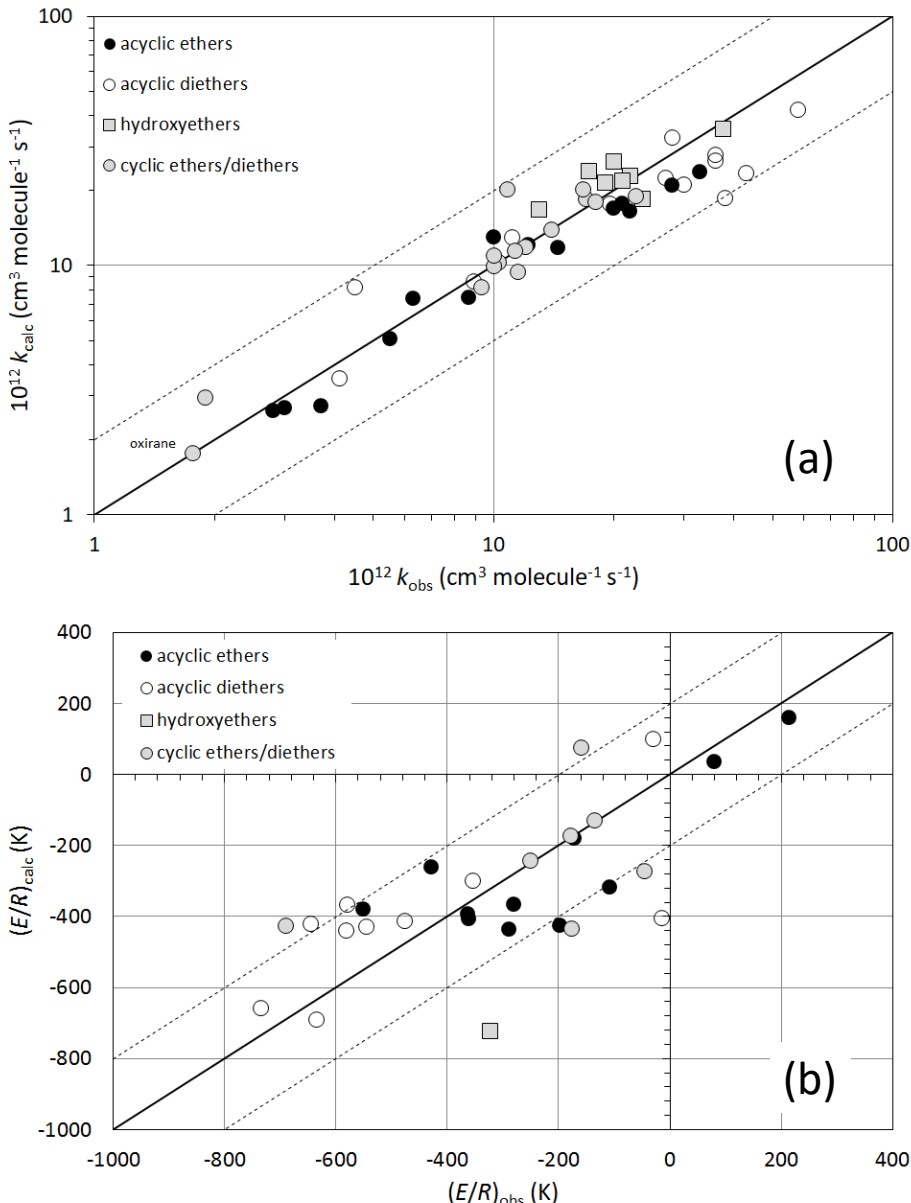

**Figure 3: (a) A log-log correlation of $k_{calc}$ and $k_{obs}$ at 298 K for saturated ethers, diethers and hydroxyethers (for presentation purposes, the value for oxirane has been scaled up by a factor of 20). The broken lines show the ± a factor of 2 range; (b) A correlation of the temperature coefficients $(E/R)_{calc}$ and $(E/R)_{obs}$ for the same compound classes. The broken lines show the ± 200 K range.**



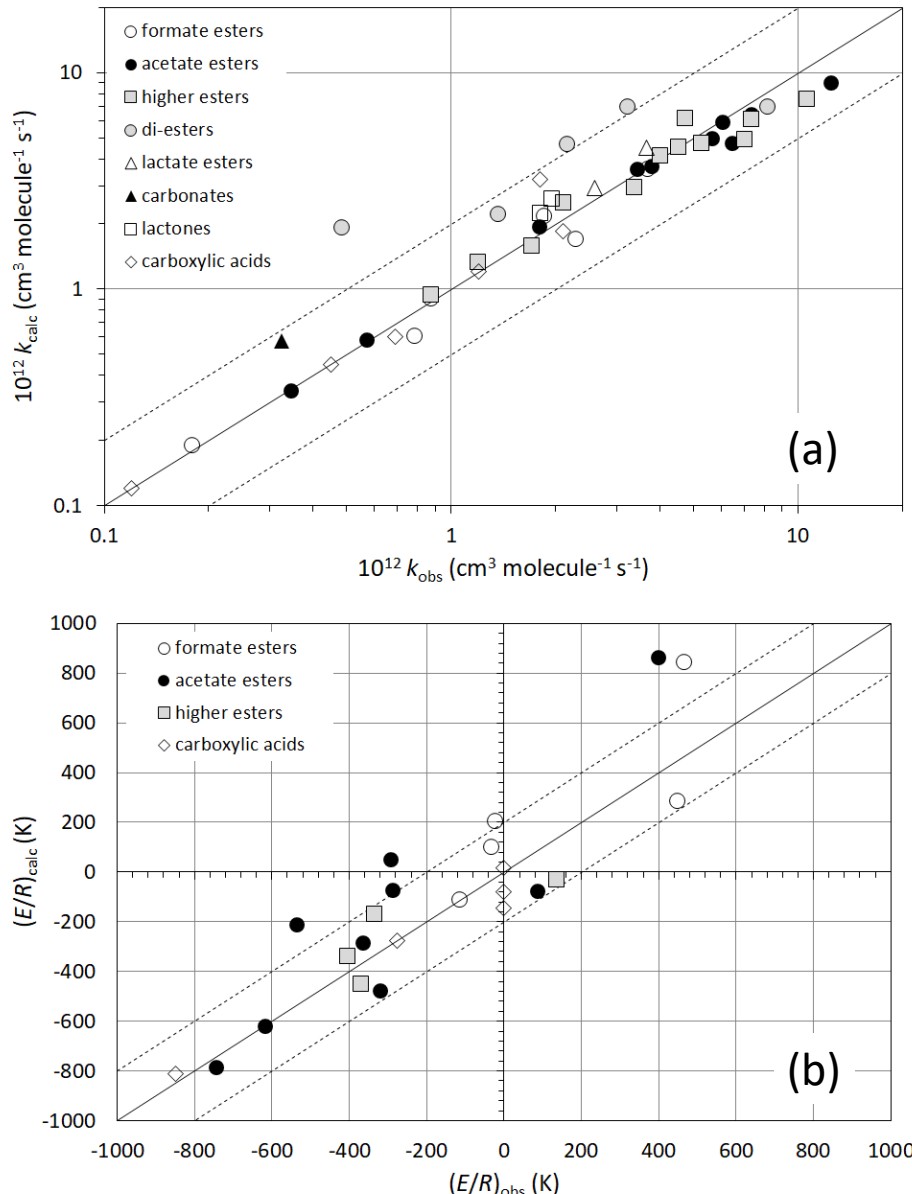

**Figure 4: (a) A log-log correlation of $k_{calc}$ and $k_{obs}$ at 298 K for saturated esters and carboxylic acids. The broken lines show the ± a factor of 2 range; (b) A correlation of the temperature coefficients $(E/R)_{calc}$ and $(E/R)_{obs}$ for the same compound classes. The broken lines show the ± 200 K range.**



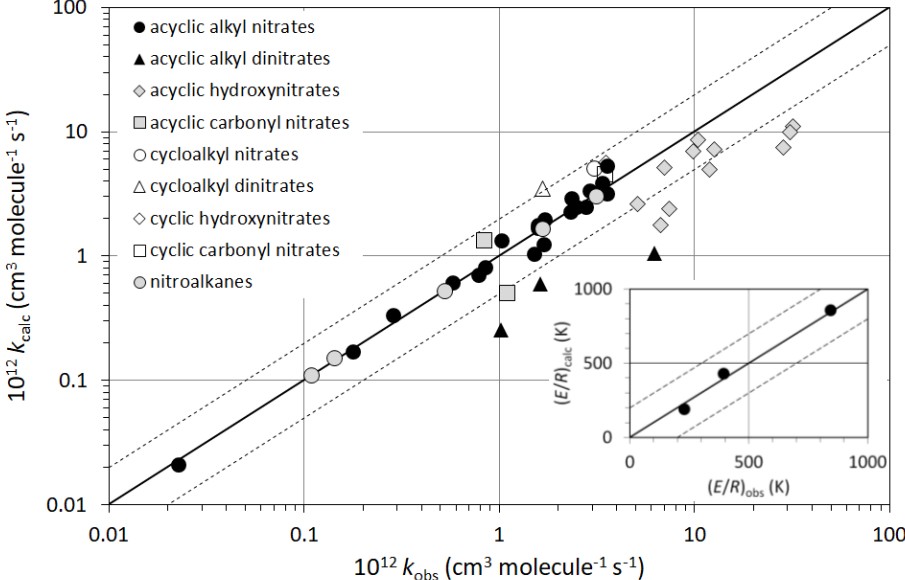

**Figure 5: A log-log correlation of $k_{calc}$ and $k_{obs}$ at 298 K for saturated organic nitrates and nitroalkanes. The broken lines show the ± a factor of 2 range. The inset plot shows a correlation of the temperature coefficients $(E/R)_{calc}$ and $(E/R)_{obs}$ for methyl-, ethyl- and 2-propyl-nitrate. The broken lines show the ± 200 K range.**



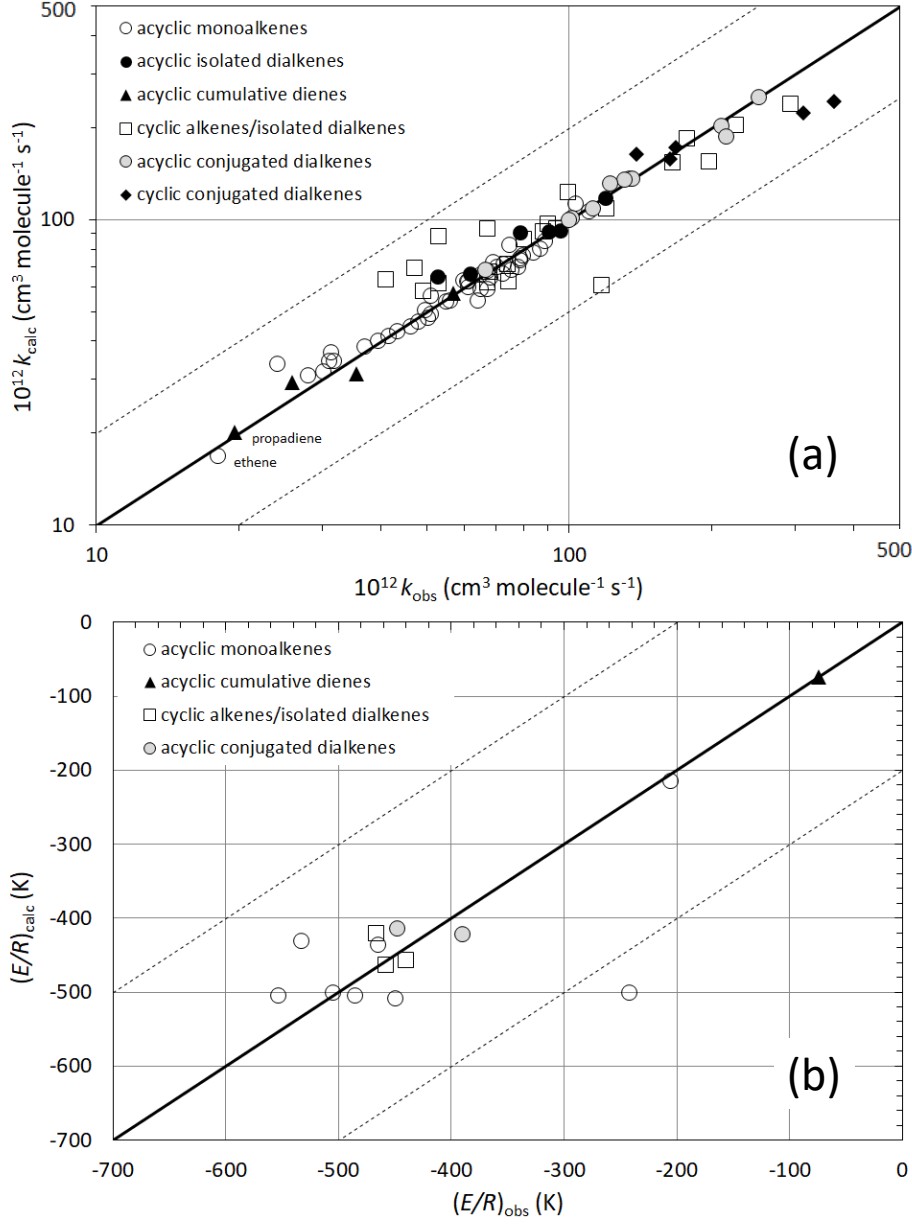

**Figure 6: (a) A log-log correlation of $k_{calc}$ and $k_{obs}$ at 298 K for alkenes (for presentation purposes, the values for ethene and propadiene have been scaled up by a factor of 2). The broken lines show the ± a factor of 2 range; (b) A correlation of the temperature coefficients $(E/R)_{calc}$ and $(E/R)_{obs}$ for the same compound classes. The broken lines show the ± 200 K range.**




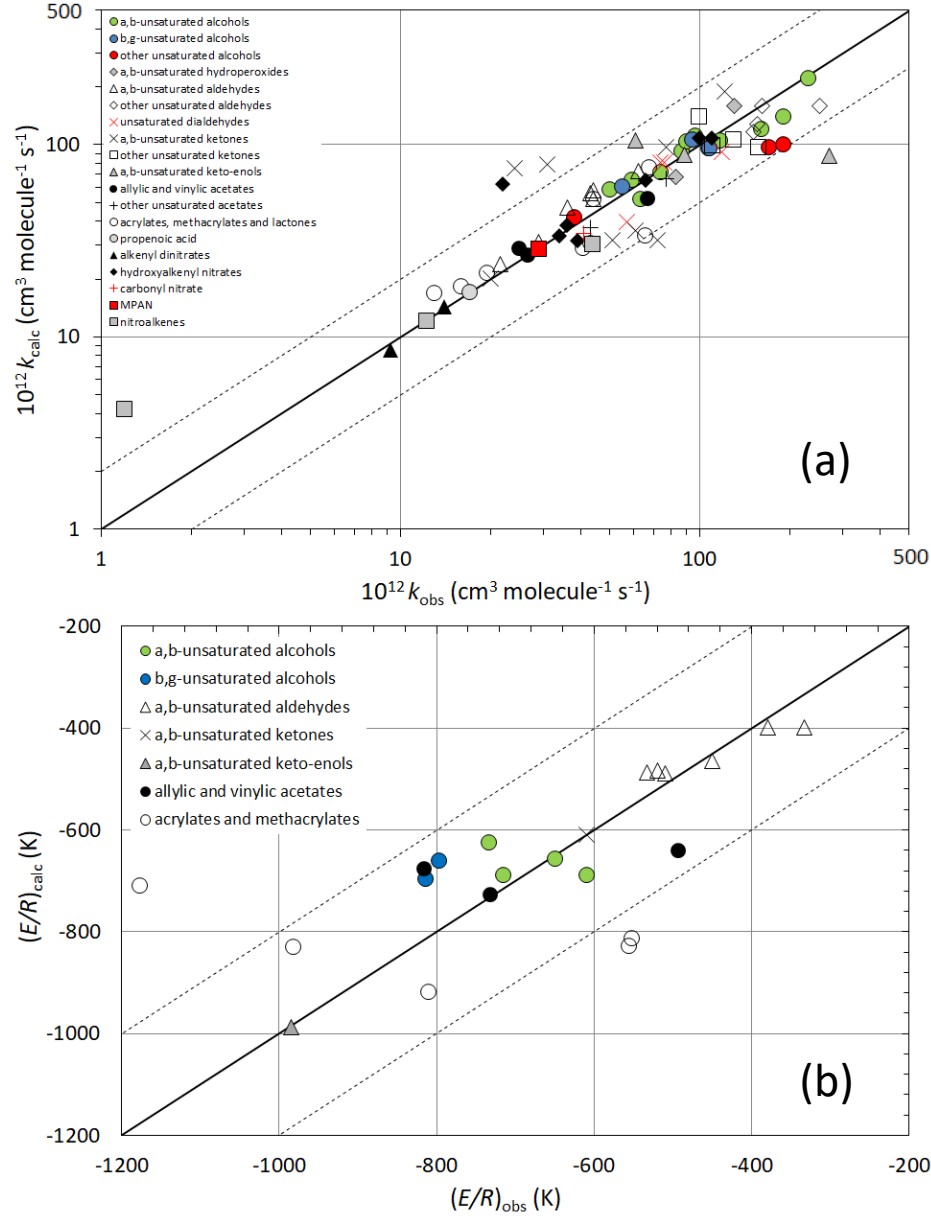

**Figure 7: A log-log correlation of $k_{calc}$ and $k_{obs}$ at 298 K for unsaturated organic oxygenates. The broken lines show the ± a factor of 2 range; (b) A correlation of the temperature coefficients $(E/R)_{calc}$ and $(E/R)_{obs}$ for the same compound classes. The broken lines show the ± 200 K range.**





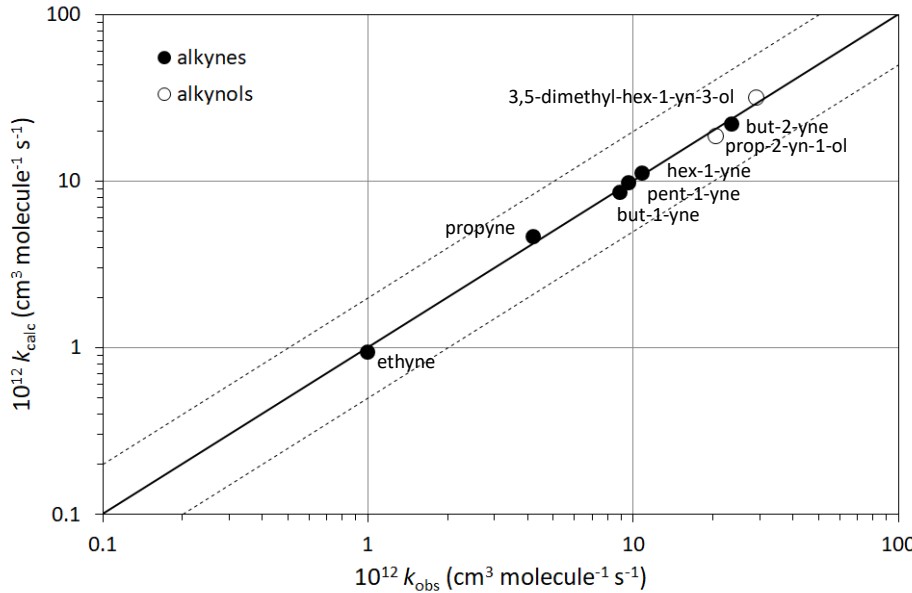

**Figure 8: A log-log correlation of $k_{calc}$ and $k_{obs}$ at 298 K for alkynes and alkynols. The broken lines show the ± a factor of 2 range.**

| | acyclic alkane | cyclic alkane | acyclic alkene | cyclic alkene | acyclic alkyne |
|---|---|---|---|---|---|
| # species | 27 | 22 | 66 | 26 | 6 |
| RMSE | 0.05 | 0.10 | 0.04 | 0.11 | 0.03 |
| MAE | 0.03 | 0.07 | 0.03 | 0.08 | 0.02 |
| MBE | −0.01 | −0.03 | 0.00 | 0.01 | 0.00 |

**Figure 9: Root mean square error, mean absolute error, mean bias error and box plot for the error distribution in the estimated log $k_{298K}$ values for various subsets of aliphatic hydrocarbons. The bottom and the top of the box are the 25th (Q1) and 75th percentile (Q3), the band is the median value. The whiskers extend to the most extreme data point which is no more than 1.5×(Q3-Q1) from the box. The points are the extrema of the distribution. The black dotted lines correspond to agreement within a factor 2.**





| | alcohol | | ketone | | ester | | nitro | unsaturated | |
| | | nitrate | | aldehyde | | ether | | saturated | |
|---|---|---|---|---|---|---|---|---|---|
| # species | 46 | 22 | 36 | 33 | 41 | 21 | 7 | 157 | 58 |
| RMSE | 0.11 | 0.09 | 0.20 | 0.07 | 0.09 | 0.09 | 0.21 | 0.11 | 0.16 |
| MAE | 0.08 | 0.07 | 0.15 | 0.05 | 0.07 | 0.07 | 0.11 | 0.07 | 0.12 |
| MBE | -0.04 | 0.01 | -0.04 | 0.02 | -0.03 | -0.03 | 0.05 | -0.03 | -0.01 |

**Figure 10: Root mean square error, mean absolute error, mean bias error and box plot for the error distribution in the estimated log $k_{298K}$ values for various subsets of monofunctional aliphatic species. The bottom and the top of the box are the 25th (Q1) and 75th percentile (Q3), the band is the median value. The whiskers extend to the most extreme data point which is no more than 1.5×(Q3-Q1) from the box. The points are the extrema of the distribution. The black dotted lines correspond to agreement within a factor 2.**

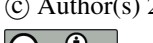



| | All data | hydro-carbon | mono-funct. | multi-funct. | saturated | unsaturated | acyclic | cyclic |
|---|---|---|---|---|---|---|---|---|
| # species | 486 | 147 | 215 | 124 | 306 | 180 | 389 | 97 |
| RMSE | 0.15 | 0.07 | 0.13 | 0.22 | 0.16 | 0.12 | 0.15 | 0.15 |
| MAE | 0.09 | 0.04 | 0.09 | 0.16 | 0.10 | 0.08 | 0.09 | 0.10 |
| MBE | −0.02 | 0.00 | −0.02 | −0.05 | −0.03 | −0.01 | −0.03 | 0.01 |

**Figure 11:** Root mean square error, mean absolute error, mean bias error and box plot for the error distribution in the estimated log $k_{298K}$ values for the full set and various subsets of aliphatic species in the database. The bottom and the top of the box are the 25th (Q1) and 75th percentile (Q3), the band is the median value. The whiskers extend to the most extreme data point which is no more than 1.5×(Q3-Q1) from the box. The points are the extrema of the distribution. The black dotted lines correspond to agreement within a factor 2.





**Table 1. Arrhenius parameters ($k = A \exp(-(E/R)/T)$ for the group rate coefficients for H atom abstraction from -CH$_3$, -CH$_2$- and -CH< groups; and the group rate coefficient values at 298 K.**

| Group | Parameter | A | $E/R$ | $k_{298K}$ |
|---|---|---|---|---|
| | | ($10^{-12}$ cm$^3$ molecule$^{-1}$ s$^{-1}$) | (K) | ($10^{-12}$ cm$^3$ molecule$^{-1}$ s$^{-1}$) |
| -CH$_3$ | $k_{prim}$ | 2.90 | 925 | 0.130 |
| -CH$_2$- | $k_{sec}$ | 4.95 | 555 | 0.769 |
| >CH- | $k_{tert}$ | 3.17 | 225 | 1.49 |

**Table 2. Substituent factors, F(X), for alkyl groups, and their temperature dependences described by $F(X) = A_{F(X)} \exp(-B_{F(X)}/T)$.**

| X | $A_{F(X)}$ | $B_{F(X)}$ | $F(X)_{298K}$ |
|---|---|---|---|
| | | (K) | |
| -CH$_3$ | 1.00 | 0 | 1.00 |
| -CH$_2$-, >CH-, >C< | 1.00 | -89 | 1.35 |

**Table 3. Ring factors, $F_{ring}$, for the reactions of OH with cyclic alkanes, and their temperature dependences described by $F_{ring} = A_{F(ring)} \exp(-B_{F(ring)}/T)$.**

| Ring (parameter) | $A_{F(ring)}$ | $B_{F(ring)}$ | $F_{ring\ (298\ K)}$ |
|---|---|---|---|
| | | (K) | |
| 3-member ring ($F_{ring}(3)$) | 0.395 | 920 | 0.018 |
| 4-member ring ($F_{ring}(4)$) | 0.634 | 130 | 0.41 |
| 5-member ring ($F_{ring}(5)$) | 0.873 | 70 | 0.69 |
| 6-member ring ($F_{ring}(6)$) | 0.95 | 0 | 0.95 |
| 7-member ring ($F_{ring}(7)$) | 1.12 | 0 | 1.12 |
| 8-member ring ($F_{ring}(8)$) | 1.16 | 0 | 1.16 |





**Table 4.** Arrhenius parameters ($k$ = A exp(-($E/R$)/T) for the rate coefficients for H-atom abstraction from hydroxy and hydroperoxy groups, and for the formyl group in RC(=O)H for various classes of R; and their values at 298 K. These values are applied independently of neighbouring group substituent factors.

| Group | Parameter | A | $E/R$ | $k_{298K}$ |
|---|---|---|---|---|
| | | ($10^{-12}$ cm$^3$ molecule$^{-1}$ s$^{-1}$) | (K) | ($10^{-12}$ cm$^3$ molecule$^{-1}$ s$^{-1}$) |
| -OH | $k_{abs(-OH)}$ | 1.28 | 660 | 0.140 |
| -OOH | $k_{abs(-OOH)}$ | 0.368 | -635 | 3.10 |
| RC(=O)H (R = -H) [a] | - | 2.70 | -135 | 4.25 |
| RC(=O)H (R = -CH$_3$) [b] | - | 4.60 | -350 | 14.9 |
| RC(=O)H (R = -CH$_2$X) [c] | $k_{abs(-CHO)n}$ | 5.08 | -420 | 20.8 |
| RC(=O)H (R = -CH(X)Y or -CH(X)(Y)Z) [d] | $k_{abs(-CHO)st}$ | 5.22 | -490 | 27.0 |
| RC(=O)H (R = -CH$_2$OX) [e] | $k_{abs(-CHO)n-\alpha O}$ | 11.7 | 140 | 7.3 |
| RC(=O)H (R = -CH(OX)Y or -C(OX)(Y)Z) [f] | $k_{abs(-CHO)st-\alpha O}$ | 11.7 [k] | -25 | 12.7 |
| RC(=O)H (R = β-hydroxyalkyl) [g] | $k_{abs(-CHO)-\beta OH}$ | 11.7 [k] | 78 | 9.0 |
| RC(=O)H (R = -C(=O)H) [h] | - | 1.55 | -340 | 4.85 |
| RC(=O)H (R = -C(=O)X) [i] | $k_{abs(-CHO)-\alpha CO}$ | 1.78 | -590 | 12.9 |
| RC(=O)H (R = >C=C<) [j] | $k_{abs(-CHO)-\alpha C=C}$ | 3.07 | -430 | 13.0 |

**Notes**

[a] Parameter is specific to formaldehyde, and is shown to illustrate trend of increasing substitution in R. Value represents rate coefficient per formyl group; [b] Parameter is specific to the formyl group in acetaldehyde, and is shown to illustrate trend of increasing substitution in R; [c] $k_{abs(-CHO)n}$ used for R = -CH$_2$X, except where X is an oxygenated group (i.e. -OX) for which $k_{abs(-CHO)n-\alpha O}$ is applied. Parameter optimized using data for aldehydes where R is an *n*-alkyl or *i*-alkyl group; [d] $k_{abs(-CHO)st}$ used for R = -CH(X)Y or -C(X)(Y)Z , except where X, Y or Z is an oxygenated group (i.e. -OX), for which $k_{abs(-CHO)st-\alpha O}$ is applied. Parameter optimized using data for aldehydes where R is a *sec*-alkyl or *tert*-alkyl group; [e] $k_{abs(-CHO)n-\alpha O}$ used for R = -CH$_2$OX. Parameter based on recommended rate coefficient for glycolaldehyde (i.e. -OX = -OH), but used as a default for aldehydes containing other oxygenated groups (e.g. -OX = -OOH, -OR, -OOR or -ONO$_2$); [f] $k_{abs(-CHO)st-\alpha O}$ used for R = -CH(OX)Y or -C(OX)(Y)Z. Parameter optimized using data for aldehydes where R is an α-hydroxyalkyl or α,β-dihydroxyalkyl group but used as a default for aldehydes containing other α-oxygenated groups (e.g. OX = -OOH, -OR, -OOR or -ONO$_2$); [g] $k_{abs(-CHO)-\beta OH}$ used for R = -CH(X)Y or -C(X)(Y)Z, when X, Y or Z = -CH$_2$OH, -CH(OH)Y' or -CH(OH)(Y')Z'. Parameter optimized using data for aldehydes where R is an β-hydroxyalkyl group; [h] Parameter is specific to glyoxal, and is shown to illustrate trend of increasing substitution in R. Value represents rate coefficient per formyl group; [i] $k_{abs(-CHO)-\alpha CO}$ used for R = -C(=O)X. Parameter based on recommended rate coefficient for methyl glyoxal; [j] $k_{abs(-CHO)-\alpha C=C}$ used for R = >C=C<, based on data for 13 α,β-unsaturated aldehydes (see Sect. 4.2); [k] In the absence of temperature-dependence studies for this class of compound, "A" is assigned the same value as for $k_{abs(-CHO)n-\alpha O}$.





**Table 5. Substituent factors, F(X), for hydroxy, hydroperoxy, peroxy and carbonyl groups, and their temperature dependences described by $F(X) = A_{F(X)} \exp(-B_{F(X)}/T)$.**

| X | $A_{F(X)}$ | $B_{F(X)}$ | $F(X)_{298K}$ |
|---|---|---|---|
| | | (K) | |
| -OH, -OOH [a], -OOR [a] | 0.497 | -590 | 3.6 |
| -CH$_2$OH, -CH(OH)-, -C(OH)< | 0.119 | -930 | 2.7 |
| -C(=O)-, -C(=O)C(=O)- | 0.309 | -350 | 1.0 |
| -CH$_2$C(=O)-, -CH(C(=O)-)-, -C(C(=O)-)< | 0.0253 | -1460 | 3.4 |

**Notes**

[a] Assumed to take the same value as F(-OH), due to limited data for compounds containing -OOH groups and no data for compounds containing -OOR groups.

**Table 6. Ring factors, $F_{ring}$, for the reactions of OH with cyclic oxygenates, and their temperature dependences described by $F_{ring} = A_{F(ring)} \exp(-B_{F(ring)}/T)$.**

| ring | cycloketones | cyclic mono-ethers | | | cyclic di-ethers | | |
|---|---|---|---|---|---|---|---|
| | $F_{ring-CO\ (298\ K)}$ | $A_{F(ring-O)}$ | $B_{F(ring-O)}$ (K) | $F_{ring-O\ (298\ K)}$ | $A_{F(ring-O')}$ | $B_{F(ring-O')}$ (K) | $F_{ring-O'\ (298\ K)}$ |
| 3-member ring | - [a] | - [b] | - [b] | 0.0079 [f] | - [a] | - [a] | - [a] |
| 4-member ring | 0.080 [b,c] | - [b] | - [b] | 0.50 [g] | - [a] | - [a] | - [a] |
| 5-member ring | 0.32 [b,d] | 1.20 | 55 | 1.00 [h] | - [b] | - [b] | 0.59 [k] |
| 6-member ring | 0.61 [b,e] | 1.59 | 290 | 0.60 [i] | 2.53 | 535 | 0.42 [l] |
| 7-member ring | - [a] | 1.61 | 190 | 0.85 [j] | 1.30 | 240 | 0.58 [m] |

**Notes**

[a] In the absence of data, $F_{ring}$ assumed to be the same as for cycloalkanes (Table 3); [b] In the absence of temperature dependence data, a value of $A_{F(ring)} = 1$ is assumed, such that $F_{ring} = \exp(298.\ln(F_{ring\ (298\ K)})/T)$; [c] Based on cyclobutanone; [d] Based on cyclopentanone; [e] Based on cyclohexanone; [f] Based on oxirane ; [g] Based on oxetane; [h] Based on tetrahydrofuran and 2-methyl-tetrahydrofuran; [i] Based on tetrahydropyran; [j] Based on oxepane; [k] Based on 1,3-dioxolane; [l] Based on 1,3-dioxane, 1,4-dioxane and 4-methyl-1,3-dioxane. Data for 1,3,5-trioxane suggest factor is also reasonable for 6-member ring cyclic tri-ethers, with the value optimized to this compound alone being 0.33; [m] Based on 1,3-dioxepane.



**Table 7.** Arrhenius parameters ($k = A \exp(-(E/R)/T)$) for the rate coefficients for H-atom abstraction from carbon atoms adjacent to oxygen linkages in mono-ethers and di-ethers; from the formyl group in formate esters, and from the carboxyl group in carboxylic acids; and their values at 298 K. These values are applied independently of neighbouring group substituent factors.

| Group | Parameter | A | E/R | $k_{298K}$ |
|---|---|---|---|---|
| | | ($10^{-12}$ cm$^3$ molecule$^{-1}$ s$^{-1}$) | (K) | ($10^{-12}$ cm$^3$ molecule$^{-1}$ s$^{-1}$) |
| -OCH$_3$ | $k_{abs(-OCH3)}$ [a] | 2.22 | 160 | 1.3 |
| -OCH$_2$R, -OCH$_2$OR, -OCH(R)<, -OCH(OR)< | $k_{abs(-OR)}$ [a,b] | 1.20 | -460 | 5.6 |
| -OCH$_2$-C-OR, -OCH(-C-OR)< (acyclic) | $k_{abs(-OCCOR)}$ [a,c] | 1.17 | -760 | 15.0 |
| ROC(=O)H | $k_{abs(ROCHO)}$ [d] | 1.70 | 910 | 0.08 |
| RC(=O)OH (R = -H) | $k_{abs(formic\ acid)}$ [e] | 0.103 | -380 | 0.37 |
| RC(=O)OH (R = alkyl) | $k_{abs(RC(O)OH)}$ [f] | 0.0287 | -880 | 0.55 |
| RC(=O)OH (R = -C(=O)-) | $k_{abs(RC(O)C(O)OH)}$ [g] | 0.0477 | -275 | 0.12 |

Notes
[a] Applies specifically to abstraction adjacent to an ethereal oxygen linkage; [b] Based on data for relevant ethers and di-ethers, and applied when R is any organic or inorganic group; [c] Applies to acyclic compounds only. Based on data for dimethoxyethane and diethoxyethane, and only applies when R is an alkyl or remotely-substituted (i.e. β or higher) alkyl group. In all other cases, $k_{abs(-OR)}$ should be applied; [d] Applies to abstraction of the formyl H-atom in formate esters and formic acid; [e] Applies to abstraction of the carboxyl H-atom in formic acid only, and is shown to illustrate trend of increasing substitution; [f] Based on data for higher alkanoic acids, but also applied as a default when R ≠ -C(=O)-; [g] Based on recommended rate coefficient for pyruvic acid, assuming reaction occurs exclusively at the carboxyl group.

5    **Table 8. Substituent factors, F(X), for oxygenated groups in ethers, esters and carboxylic acids, and their temperature dependences described by $F(X) = A_{F(X)} \exp(-B_{F(X)}/T)$.**

| X | $A_{F(X)}$ | $B_{F(X)}$ | $F(X)_{298K}$ |
|---|---|---|---|
| | | (K) | |
| -CH$_2$OR, -CH(OR)-, -C(OR)< | 0.122 | -1000 | 3.5 |
| -OC(=O)H | 0.0251 | -1050 | 0.85 |
| -OC(=O)R | 0.0310 | -1270 | 2.2 |
| -CH$_2$C(=O)OR, -CH(C(=O)OR)-, -C(C(=O)OR)< [a] | 0.0215 | -1440 | 2.7 |
| -C(=O)OR [a] | 0.783 | 200 | 0.4 |
| -C(=O)C(=O)OH, -C(=O)C(=O)OR [b] | 0 | 0 | 0 |

Notes
[a] Also applied to carboxylic acids (i.e. for -OR = -OH); [b] Based on recommended rate coefficient for pyruvic acid, assuming reaction occurs exclusively at the carboxyl group.





**Table 9. Substituent factors, F(X), for oxidized nitrogen groups, and their temperature dependences described by $F(X) = A_{F(X)} \exp(-B_{F(X)}/T)$.**

| X | $A_{F(X)}$ | $B_{F(X)}$ | $F(X)_{298K}$ |
|---|---|---|---|
| | | (K) | |
| $-ONO_2$ [a] | 0.127 | -70 | 0.16 |
| $-CH_2ONO_2$, $-CH(ONO_2)-$, $-C(ONO_2)<$ [a] | 0.0397 | -640 | 0.34 |
| $-NO_2$ [b,c] | - | - | 0 |
| $-CH_2NO_2$, $-CH(NO_2)-$, $-C(NO_2)<$ [b,c] | - | - | 0.31 |
| $-C(=O)OONO_2$ [c,d] | - | - | 0.1 |

**Notes**

[a] Based on data for acyclic alkyl nitrates; [b] Based on atmospheric pressure data for nitroalkanes, with an addition component to the reaction, $k_{add(-NO2)} = 1.1 \times 10^{-13}$ cm$^3$ molecule$^{-1}$ s$^{-1}$, optimized simultaneously (see Sect. 3.2.5); [c] Parameters should provisionally be assumed to be temperature independent; [d] Set so that $k_{calc}$ is ≈ 50 % of the upper limit reported by Talukdar et al. (1995);



**Table 10. Arrhenius parameters ($k = A \exp(-(E/R)/T)$ for the group rate coefficients for OH addition to C=C bonds in monoalkenes and polyalkenes; and the group rate coefficient values at 298 K.[a]**

| Group | Parameter | A | E/R | $k_{298K}$ |
|---|---|---|---|---|
| | | ($10^{-12}$ cm$^3$ molecule$^{-1}$ s$^{-1}$) | (K) | ($10^{-12}$ cm$^3$ molecule$^{-1}$ s$^{-1}$) |
| *isolated C=C bonds* | | | | |
| -C=CH$_2$ | $k_{prim\text{-}add}$ | **2.04** | **-215** | **4.2** |
| -C=CHCH$_3$ | $k_{sec\text{-}add}$ | **4.30** | **-540** | **26.3** |
| -C=C(CH$_3$)$_2$ | $k_{tert\text{-}add}$ | **8.13** | **-550** | **51.5** |
| *conjugated C=C-C=C bonds* [b] | | | | |
| -C=CHC=CH$_2$ | $k_{sec,prim}$ | **6.74** | **-445** | **30** |
| -C=C(CH$_3$)-C=CH$_2$ | $k_{tert,prim}$ | **13.70** | **-445** | **61** |
| -C=CH-C=CHCH$_3$ | $k_{sec,sec}$ | 8.99 | -445 [c] | **40** |
| -C=C(CH$_3$)-C=CHCH$_3$ | $k_{tert,sec}$ | 16.62 | -445 [c] | **74** |
| -C=CH-C=C(CH$_3$)$_2$ | $k_{sec,tert}$ | 10.56 | -445 [c] | **47** |
| -C=C(CH$_3$)-C=C(CH$_3$)$_2$ | $k_{tert,tert}$ | 22.24 | -445 [c] | **99** [d] |
| *cumulative C=C=C bonds* | | | | |
| -C=C=C | $k_v$ | **0.777** | **-75** | **1.0** |
| -C(=CH$_2$)$_2$ | $k_{pp}$ | **6.22** | **-75** | **8.0** |
| -C(=CH$_2$)=CHCH$_3$ | $k_{ps}$ | 5.13 | -495 [e] | **27** |
| -C(=CH$_2$)=C(CH$_3$)$_2$ | $k_{pt}$ | 9.27 | -525 [e] | **54** |
| -C(=CHCH$_3$)=CHCH$_3$ | $k_{ss}$ | 8.17 | -540 [e] | 50 [f] |
| -C(=CHCH$_3$)=C(CH$_3$)$_2$ | $k_{st}$ | 12.04 | -545 [e] | 75 [g] |
| -C(=C(CH$_3$)$_2$)=C(CH$_3$)$_2$ | $k_{tt}$ | 15.79 | -550 [e] | 100 [h] |

**Notes**

[a] Reference parameters are defined for degrees of substitution by -CH$_3$ (see Sect. 4.1). Values shown in bold were optimized by the procedures described in Sect. 4.1. Other values could not be fitted, owing to insufficient data, but were estimated as described in the following notes; [b] Product radicals are assumed to be formed 50% $E$- and 50% $Z$- unless specific information is available; [c] Assumed equal to that optimized for $k_{sec,prim}$ and $k_{tert,prim}$; [d] Estimated value, unchanged from Peeters et al. (2007); [e] $E/R$ values based on the weighted average of those for the corresponding combinations of $k_{prim\text{-}add}$, $k_{sec\text{-}add}$ and $k_{tert\text{-}add}$; [f] $k_{ss}$ estimated to be $\approx$ $2k_{sec\text{-}add}$ at 298 K; [g] $k_{st}$ estimated to be $\approx k_{sec\text{-}add} + k_{tert\text{-}add}$ at 298 K; [h] $k_{tt}$ estimated to be $\approx 2k_{tert\text{-}add}$ at 298 K (see Sect. 4.1.6).




**Table 11. Substituent factors, F'(X) = $A_{F'(X)}$ exp(-$B_{F'(X)}$/T), for the addition OH to C=C bonds [a].**

| X | $A_{F'(X)}$ | $B_{F'(X)}$ | $F'(X)_{298K}$ |
|---|---|---|---|
| | | (K) | |
| -$C_nH_{2n+1}$ (acyclic linear alkyl) [b] | - | - | 1 + 0.14[1-exp(-0.35($C_n$-1))] |
| other alkyl and alkenyl (and default) [c] | 1.0 | 0 | 1.0 |
| -OH [d] | 0.249 | -515 | 1.4 |
| -$CH_2OH$, -CH(OH)-, -C(OH)<, -C-$CH_2OH$, -C-CH(OH)-, -C-C(OH)<, -$CH_2OOH$, -CH(OOH)-, -C(OOH)< [e] | 0.951 | -190 | 1.8 |
| -C(=O)H [f] | 0.423 | 145 | 0.26 |
| -C(=O)- [g] | 0.328 | -180 | 0.6 |
| -C(=O)OH, -C(=O)OR [h] | 0.094 | -480 | 0.47 |
| -OC(=O)R [i] | 0.508 | -180 | 0.93 |
| -C-OC(=O)R [j] | 0.319 | -230 | 0.69 |
| -$CH_2ONO_2$, -CH($ONO_2$)-, -C($ONO_2$)< [k,l] | - | - | 0.26 |
| -C-$CH_2ONO_2$, -C-CH($ONO_2$)-, -C-C($ONO_2$)< [l,m] | - | - | 0.6 |
| -C(=O)$OONO_2$ [l,n] | - | - | 0.47 |
| -$NO_2$ [o] | - | - | 0.0 |
| -$CH_2NO_2$, -CH($NO_2$)-, -C($NO_2$)< [l,p] | - | - | 0.3 |

**Notes**

[a] F'(X) quantifies the effect of replacing a -$CH_3$ substituent by the given group; [b] Based on results of Aschmann and Atkinson (2008) and Nishino et al. (2009), and applied to acyclic linear alkyl groups only. Results in enhancements of up to 14 % and can be ignored to a first approximation. Assumed to be temperature independent; [c] Also used as a default for groups with remote substituents; [d] Based on limited information (three hydroxy ketones), and primarily optimized using temperature-dependent data for 4-hydroxy-pent-3-en-2-one; [e] Primarily based on data for ten α,β-unsaturated (allylic) alcohols (four temperature-dependent) and five β,γ-unsaturated alcohols (two temperature-dependent), but also taking account of data for multifunctional compounds containing hydroxyl groups. Also assumed to apply to α,β-unsaturated hydroperoxides, based on room temperature data of St. Clair et al. (2016); [f] Primarily based on data for seven α,β-unsaturated aldehydes (six temperature-dependent) and six α,β-unsaturated dialdehydes (none temperature-dependent); [g] Data do not give a well-defined value. Assigned factor is based on temperature-dependent data for methylvinyl ketone, the most studied compound in this class; [h] Based on room temperature data for propenoic acid and data for six acrylate and methacrylate esters (five temperature-dependent); [i] Based on temperature-dependent data for vinyl acetate and i-propenyl acetate ; [j] Based on temperature-dependent data for allyl acetate; [k] Based on room temperature data for two α,β-unsaturated dinitrates, four α,β-unsaturated hydroxynitrates and one α,β-unsaturated nitro-oxy aldehyde; [l] F'(X) should provisionally be assumed to be temperature independent; [m] Based on room temperature data for three β,γ-unsaturated dinitrates; [n] Based on room temperature data for MPAN; [o] Based on room temperature data for nitroethene and 1-nitrocyclohexene (N.B. $k_{add(-NO2)}$ assumed to take a value of zero for 1-nitroalkenes to avoid formation of a vinyl radical product); [p] Based on room temperature data for 3-nitropropene.





**Table 12. Substituent factors F(X) related to the H-atom abstraction reactions of OH adjacent to C=C bonds [a], and their temperature dependences described by $F(X) = \exp(-B_F(X)/T)$.**

| X | $F(X)_{298 K}$ | $B_{F(X)}$ (K) |
|---|---|---|
| -C=CH$_2$ | **2.5** | -275 |
| -C=CHR | **6.2** | -545 |
| -C=CR$_2$ | **6.2** | -545 |
| -C=C=C< | 1.0 [b] | 0 |
| (-C=CH$_2$)$_2$ | 5.0 [c,d] | -480 |
| -C=CH-C=CH$_2$, (-C=CH$_2$)(-C=CHR) | 8.7 [c,e] | -645 |
| -C=C(R)-C=CH$_2$, (-C=CH$_2$)(-C=CR$_2$) | 8.7 [c,f] | -645 |
| -C=CH-C=CHR, (-C=CHR)$_2$ | 12.4 [c,g] | -750 |
| -C=C(R)-C=CHR, -C=CH-C=CR$_2$, (-C=CHR)(-C=CR$_2$) | 12.4 [c,h] | -750 |
| -C=C(R)-C=CR$_2$, (-C=CR$_2$)$_2$ | 12.4 [c,i] | -750 |

**Notes**

[a] R denotes any alkyl group. Values shown in bold were optimized or assigned by the procedures described in Sect. 4.1. Other values could not be fitted owing to insufficient data, but were estimated as described in Sect. 4.1 and the following notes; [b] Assumed to have no activating influence because the resonant radical possesses partial vinyl character; [c] Substituent factors related to formation of "superallyl" resonant structures assumed equal to the corresponding sum of those for formation of the component allyl structures; [d] Assumed equal to 2F(-C=CH$_2$); [e] Assumed equal to F(-C=CH$_2$) + F(-C=CHR); [f] Assumed equal to F(-C=CH$_2$) + F(-C=CR$_2$); [g] Assumed equal to 2F(-C=CHR); [h] Assumed equal to F(-C=CHR) + F(-C=CR$_2$); [i] Assumed equal to 2F(-C=CR$_2$).



**Table 13. Comparison of estimated and observed total branching ratios for H-atom abstraction, $k_{abs}/(k_{abs} + k_{add})$. The estimated values correspond to 298 K, and the observed values are for temperatures at or near 298 K.**

| Compound | Branching ratio | | Comment |
|---|---|---|---|
| | calculated | observed | |
| propene | 1.0 % | < 2 % | (a) |
| but-1-ene | 6.1 % | < 10 % | (b) |
| | | < 10 % | (c) |
| | | (8 ± 3) % | (a) |
| | | (5 ± 2) % | (d) |
| *trans*-but-2-ene | 3.0 % | (3 ± 1) % | (a) |
| cyclohexa-1,3-diene | 15.6 % | 8.9 % | (e) |
| | | (15 ± 6) % | (f) |
| | | < 10 % | (g) |
| | | (8.1 ± 0.2) % | (h) |
| cyclohexa-1,4-diene | 14.7 % | (15.3 ± 0.3) % | (e) |
| | | (26 ± 9) % | (f) |
| | | (12.5 ± 1.2) % | (i) |
| limonene | 13.7 % | (34 ± 8) % | (j) |
| α-phellandrene | 21.3 % | (28 ± 7) % | (f) |
| | | (27 ± 10) % | (g) |
| α-terpinene | 17.9 % | (30 ± 8) % | (f) |
| | | (30 ± 7) % | (g) |
| γ-terpinene | 15.5 % | (13.6 ± 2.5) % | (k) |
| | | (31 ± 9) % | (j) |

**Comments**

Observed values reported in the following studies: [a] Loison et al. (2010); [b] Hoyermann and Sievert (1983); [c] Atkinson et al. (1985); [d] Loison et al. (2010) re-evaluation of Biermann et al. (1982); [e] Ohta (1984); [f] Peeters et al. (1999a); [g] Peeters et al. (1999b); [h] Jenkin et al. (2005); [i] Tuazon et al. (2003); [j] Rio et al. (2010); [k] Aschmann et al. (2011).



**Table 14. Representative rate coefficients for prompt decomposition/ring-opening reactions of thermalized organic radicals ($k_{dec}$), relative to those for addition of $O_2$ ($k_{O2}$) for primary, secondary and tertiary radicals [a].**

| Radical | Product(s) | $k_{dec}/k_{O2}$ | Relative rate [b] | comment |
|---|---|---|---|---|
| $>\dot{C}$-OOH<br>$>\dot{C}$-OOR<br>$>\dot{C}$-ONO$_2$ | $>C=O + OH$<br>$>C=O + RO$<br>$>C=O + NO_2$ | - | - | (c) |
| $-C(=O)\dot{C}=O$ | $-\dot{C}=O + CO$ | $2.35 \times 10^{21} \exp(-1405/T)$ | 4.1 | (d) |
| (1) | (2) | (2) = primary radical<br>$8.9 \times 10^{23} \exp(-3445/T)$, (1) = primary<br>$4.2 \times 10^{23} \exp(-3445/T)$, (1) = secondary<br>$3.1 \times 10^{23} \exp(-3445/T)$, (1) = tertiary | 1.7<br>0.79<br>0.58 | (e),(f) |
|  |  | (2) = secondary radical<br>$8.9 \times 10^{23} \exp(-2905/T)$, (1) = primary<br>$4.2 \times 10^{23} \exp(-2905/T)$, (1) = secondary<br>$3.1 \times 10^{23} \exp(-2905/T)$, (1) = tertiary | 10<br>4.8<br>3.5 | (e),(f) |
|  |  | (2) = tertiary radical<br>$8.9 \times 10^{23} \exp(-2510/T)$, (1) = primary<br>$4.2 \times 10^{23} \exp(-2510/T)$, (1) = secondary<br>$3.1 \times 10^{23} \exp(-2510/T)$, (1) = tertiary | 38<br>19<br>13 | (e),(f) |
| (1) | | $10.4 \times 10^{23} \exp(-2200/T)$, (1) = primary<br>$4.9 \times 10^{23} \exp(-2200/T)$, (1) = secondary<br>$3.6 \times 10^{23} \exp(-2200/T)$, (1) = tertiary | 130<br>60<br>44 | (g),(f) |
|  |  | uncompetitive | $< 10^{-3}$ | (g) |

**Comments**

[a] Rate coefficients adopted for reactions of primary, secondary and tertiary alkyl peroxy radicals with $O_2$ ($k_{O2}$, in units $10^{-12}$ cm$^3$ molecule$^{-1}$ s$^{-1}$) are 8, 17 and 23, respectively; based on the (high pressure limit values) reported by Lenhardt et al. (1990) for 1-butyl, 2-butyl and 2-methyl-2-propyl radicals. These values are expected to have a weak temperature dependence, and are assumed here to be independent of temperature over the tropospheric range;

[b] Illustrative value of $k_{dec}/k_{O2}[O_2]$ for air at 298 K and 760 Torr;

[c] These processes are estimated to occur spontaneously on the picosecond timescale (Vereecken et al., 2004; Vereecken, 2008) and can be assumed to occur exclusively for all relevant organic radicals;

[d] Based on the relative rate coefficient reported by Jagiella and Zabel (2008) for thermalized CH$_3$C(=O)$\dot{C}$=O radicals. N.B. chemically activated [R'C(=O)$\dot{C}$=O]$^{\ddagger}$ radicals, formed specifically from the reactions of OH with R'C(=O)CHO, are assumed to decompose exclusively to R'$\dot{C}$O + CO (60 %) and R' + 2 CO (40 %), based on the observations of Baeza-Romero et al. (2007) for the reaction of OH with methyl glyoxal (CH$_3$C(=O)CHO) (see Sect. 6.1);

[e] Based on the average of rate coefficients reported by Bowry (1991) for a series of cyclopropyl-alkyl radicals, representing the value per relevant bond;

[f] The values of $k_{dec}/k_{O2}$ shown for the secondary and tertiary reagent radical (1) can be adjusted approximately for the effects of a substituent group, X, in *cyclo*-propyl-$\dot{C}$H-X and *cyclo*-propyl-$\dot{C}$(R')-X, using the following temperature-independent factors: (i) $F_{dec/O2}$(-OH) = 0.6, based on rate coefficients reported for reactions of $O_2$ with the $\alpha$-hydroxyalkyl radicals, CH$_3\dot{C}$HOH (http://iupac.pole-ether.fr/), C$_2$H$_5\dot{C}$HOH (Miyoshi et al., 1990) and CH$_3\dot{C}$(OH)CH$_3$ (Miyoshi et al., 1990); (ii) $F_{dec/O2}$(-C(=O)-) = 7.0, based on rate coefficients reported for reactions of $O_2$ with the $\beta$-oxoalkyl/vinoxy radicals CH$_3$C(=O)$\dot{C}$H$_2$ (http://iupac.pole-ether.fr/) and CH$_3\dot{C}$HC(=O)H (Oguchi et al., 2001); (iii) $F_{dec/O2}$(-C(OH)<) = 1.7, based on rate coefficients reported for reactions of $O_2$ with the $\beta$-hydroxyalkyl radicals CH$_3\dot{C}$HCH$_2$OH and CH$_3$CH(OH)$\dot{C}$H$_2$ (Miyoshi et al., 1990); (iv) $F_{dec/O2}$(=O) = 4.5, based on rate coefficient reported for reaction of $O_2$ with CH$_3\dot{C}$O (http://iupac.pole-ether.fr/); and (v) $F_{dec/O2}$(=C<) = 0.9, based on rate coefficient reported for reaction of $O_2$ with CH$_2$=$\dot{C}$H (Matsugi and Miyoshi, 2014). For other substituents, $F_{dec/O2}$ = 1.0 is assumed, in the absence of data;

[g] Based on the rate coefficients calculated for the oxiranyl-methyl radical in the theoretical study of Smith et al. (1998). The rate coefficient for C-O bond breaking agrees with the lower limit value at 343 K, reported by Krosley and Gleicher (1993).



**Table 15.** Prompt rearrangements of chemically activated [ROO]$^{\ddagger}$ adducts, formed from the reactions of organic radicals with $O_2$.

| Radical | Products | comment |
|---|---|---|
| >ĊOH | [>C(OH)OO]$^{\ddagger}$ → >C=O + HO$_2$ (1-β) <br> → >C(OH)O$_2$ (β) | (a) |
| >C=ĊR' | [>C=C(R')OO]$^{\ddagger}$ → >C=O + R' + CO (65%) <br> → >C=O + R'Ċ=O (35%) | (b) |
| -C(OH)=ĊR' | [-C(OH)=C(R')OO]$^{\ddagger}$ → -C(=O)C(=O)R' + OH (70 %) <br> → -C(=O)OH + R' + CO (30 %) | (c) |
| 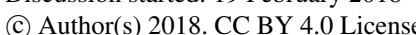 | 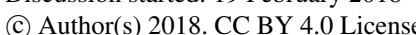 + HO$_2$ | (d) |

**Comments**

[a] The fractional formation of thermalized >C(OH)O$_2$ radicals (β) is defined in terms of the size of the organic group, >COH (see Table 16, Fig. S10 and Sect. 6.2);

[b] Product channels reported for the reaction of O$_2$ with vinyl radicals (e.g. Carpenter, 1995; Eskola and Timonen, 2003; Matsugi and Miyoshi, 2014). Product ratios applied here are based on observations for the reaction of O$_2$ with the methylvinyl radical (Orlando et al., 1999), formed during the OH-initiated oxidation of methacrolein (see Sect. 6.2);

[c] Product channels reported for the reaction of O$_2$ with 2-hydroxyvinyl radicals, formed from the addition of OH to alkynes (e.g. Hatakeyama et al., 1986; Yeung et al., 2005; Lockhart et al., 2013). Product ratios applied here are based on OH yields reported by Lockhart et al. (2013) for ethyne, propyne and but-1-yne, and informed by the observations of α-dicarbonyls (-C(=O)C(=O)R') and carboxylic acids (-C(=O)OH) reported by Hatakeyama et al. (1986) and Yeung et al. (2005) (see Sect. 6.2);

[d] Applies generally to cyclohexadienyl and alkyl-substituted cyclohexadienyl radicals (see Sect. 6.2). Products are based on the reported formation of aromatic hydrocarbon products in a number of studies (Ohta et al., 1984; Tuazon et al., 2003; Jenkin et al., 2005; Aschmann et al., 2011). Reaction may proceed either via formation of an [ROO]$^{\ddagger}$ adduct, or via a direct H atom abstraction mechanism.



**Table 16. Reported fractional formation of thermalized α-hydroxy peroxy radicals (β) from the reactions of α-hydroxy radicals with O$_2$.**

| Radical | $n_{CON}$ | Thermalization fraction (β) | Comment |
|---|---|---|---|
| ĊH$_2$OH | 2 | 0 | (a) |
| CH$_3$ĊHOH | 3 | 0 | (b) |
| (CH$_3$)$_2$ĊOH | 4 | 0 | (b) |
| (cyclohexanol structure)—OH | 7 | 0.016-0.077 | (b) |
| CH$_3$(CH$_2$)$_5$ĊHOH | 8 | 0.13 | (c) |
| CH(=O)CH=CH(CH$_2$)$_2$ĊHOH | 8 | (0.50 ± 0.25) | (d) |
| (structure) | 12 | 0.75 | (e) |
| (structure) | 12 | 0.9 | (e) |

**Comments**

[a] Based on the theoretical studies of Dibble (2002) and Hermans et al. (2005) [b] Based on the theoretical study of Hermans et al. (2005), with support from the laboratory observations of the OH initiated oxidation of small alkenes and alcohols in the presence of NO (e.g. Niki et al., 1978; Carter et al., 1979); [c] Based on HCOOH formation during the OH-initiated oxidation of 7-tetradecene in the presence of NO, reported by Aschmann et al. (2010); [d] Based on HCOOH formation during the OH-initiated oxidation of cyclohexa-1,3-diene in the presence of NO, reported by Jenkin et al. (2005). An approximate value of β was extracted from simulation of a complex system, and the wide error bars are assigned here, based on comments in Jenkin et al. (2005); [e] Based on the theoretical study of OH-initiated α-pinene oxidation by Capouet et al. (2004), with support from the laboratory observations of HCOOH formation during the OH-initiated oxidation of a series of monoterpenes in the presence of NO (Orlando et al., 2000).



**Table 17. Partial rate coefficients for the addition of $O_2$ to radicals possessing an allyl resonance.[a,b]**

| Reaction | $k_f$ | Comment |
|---|---|---|
| | $(10^{-13} \text{ cm}^3 \text{ molecule}^{-1} \text{ s}^{-1})$ | |
| *unsubstituted* [c] | | |
| $-C=C-\dot{C}H_2 + O_2 \rightarrow -C=C-CH_2O_2$ | 3 | (e) |
| $-C=C-\dot{C}HR + O_2 \rightarrow -C=C-CH(O_2)R'$ | 10 | (f) |
| $-C=C-\dot{C}R_2 + O_2 \rightarrow -C=C-C(O_2)R'_2$ | 10 | (g) |
| *β-/δ-hydroxy substituted* [c,d] | | |
| $(E)$ $-C(OH)-C=C-\dot{C}H_2 + O_2 \rightarrow (E)$ $-C(OH)-C=C-CH_2O_2$ | 5 | (h),(i) |
| $(Z)$ $-C(OH)-C=C-\dot{C}H_2 + O_2 \rightarrow (Z)$ $-C(OH)-C=C-CH_2O_2$ | 26 | (h),(j) |
| $(E \text{ or } Z)$ $-C=C-\dot{C}H-C(OH)- + O_2 \rightarrow -C=C-CH(O_2)-C(OH)-$ | 35 | (h),(k) |
| $(E \text{ or } Z)$ $-C=C-\dot{C}(R')-C(OH)- + O_2 \rightarrow -C=C-C(O_2)(R')-C(OH)-$ | 30 | (h),(l) |

**Comments**

[a] Rate coefficients are high pressure limits and are assumed to be independent of temperature over the relevant atmospheric range; [b] Each partial rate coefficient represents addition of $O_2$ at one of two possible sites in a given allyl radical (or of three possible sites in a superallyl radical); [c] Unspecified substituents are either H atoms or alkyl groups, but the parameters are also used a defaults for organic groups containing remote oxygenated substituents for which no information is available (see Sect. 6.3); [d] Formed specifically from the addition of OH to conjugated dialkene structures; [e] Based on a reported total rate coefficient of $6 \times 10^{-13}$ cm$^3$ molecule$^{-1}$ s$^{-1}$ for $CH_2CHCH_2 + O_2$ (Jenkin et al., 1993; Rissanen et al., 2012); [f] Based on reported total rate coefficients of $6 \times 10^{-13}$ cm$^3$ molecule$^{-1}$ s$^{-1}$ for $CH_2CHCH_2 + O_2$ (Jenkin et al., 1993; Rissanen et al., 2012) and $1.3 \times 10^{-12}$ cm$^3$ molecule$^{-1}$ s$^{-1}$ for $CH_3CHCHCH_2 + O_2$ (Knyazev and Slagle, 1998); [g] Assumed equivalent to rate coefficient for $-C=C-\dot{C}HR + O_2 \rightarrow -C=C-CH(O_2)R$; [h] Based on a geometric average of rate coefficients calculated for (or assigned to) relevant structures formed from addition of OH to isoprene. As recommended by Peeters (2015), these are based on those calculated by Peeters et al. (2014), but with each increased by a factor of 5 on the basis of the experimental characterization of the equilibration of peroxy radicals in each subset, as reported in preliminary form by Crounse et al. (2014) and applied in MCM v3.3.1; see Table S2 for further details; [i] Based on the 298 K rate coefficients for trans-1-OH + $O_2 \rightarrow$ E-1-OH-4-OO and trans-4-OH + $O_2 \rightarrow$ E-4-OH-1-OO (see Table S2), and also applied to corresponding secondary and tertiary radicals in the absence of data; [j] Based on rate coefficients for cis-1-OH + $O_2 \rightarrow$ Z-1-OH-4-OO and cis-4-OH + $O_2 \rightarrow$ Z-4-OH-1-OO (see Table S2), and also applied to corresponding secondary and tertiary radicals in the absence of data; [k] Based on rate coefficients for cis-4-OH + $O_2 \rightarrow$ 4-OH-3-OO and trans-4-OH + $O_2 \rightarrow$ 4-OH-3-OO (see Table S2); [l] Based on rate coefficients for cis-1-OH + $O_2 \rightarrow$ 1-OH-2-OO and trans-1-OH + $O_2 \rightarrow$ 1-OH-2-OO (see Table S2).



**Table 18. Arrhenius parameters ($k_r = A_r \exp(-(E/R)_r/T)$ for the rate coefficients for the decomposition of allyl peroxy radicals; and the rate coefficient values at 298 K.[a]**

| Reaction | $A_r$ | $(E/R)_r$ | $k_{r\,298K}$ | Comment |
|---|---|---|---|---|
| | $(10^{14}\ s^{-1})$ | (K) | $(s^{-1})$ | |
| *unsubstituted* [b] | | | | |
| -C=C-CH$_2$O$_2$ → -C=C-ĊH$_2$ + O$_2$ | 0.16 | 8900 | 1.7 | (d),(e) |
| -C=C-CH(O$_2$)R' → -C=C-ĊHR' + O$_2$ | 1.6 | 9610 | 1.6 | (d),(f) |
| -C=C-C(O$_2$)R'$_2$ → -C=C-ĊR'$_2$ + O$_2$ | 1.6 | 9610 | 1.6 | (g) |
| *β-/δ-hydroxy substituted* [c] | | | | |
| (*E*) -C(OH)-C=C-CH$_2$O$_2$ → (*E*) -C(OH)-C=C-ĊH$_2$ + O$_2$ | 2.5 | 9510 | 3.5 | (h),(i) |
| (*Z*) -C(OH)-C=C-CH$_2$O$_2$ → (*Z*) -C(OH)-C=C-ĊH$_2$ + O$_2$ | 40 | 10050 | 9.0 | (h),(j) |
| -C=C-CH(O$_2$)-C(OH)- → (*E*) -C=C-ĊH-C(OH)- + O$_2$  (50 %)   → (*Z*) -C=C-ĊH-C(OH)- + O$_2$  (50 %) | 210 | 11640 | 0.23 | (h),(k) |
| -C=C-C(R')(O$_2$)-C(OH)- → (*E*) -C=C-Ċ(R')-C(OH)- + O$_2$  (50 %)   → (*Z*) -C=C-Ċ(R')-C(OH)- + O$_2$  (50 %) | 170 | 11030 | 1.4 | (h),(l) |

**Comments**

[a] Rate coefficients are high pressure limits. Parameters are also assumed to apply to superallyl peroxy radicals; [b] Unspecified substituents are either H atoms or alkyl groups, but the parameters are also used a defaults for organic groups containing remote oxygenated substituents for which no information is available (see Sect. 6.3); [c] Formed specifically from the addition of OH and O$_2$ to conjugated dialkene structures; [d] Determined from the expression $k_r = k_f/K$, where K is the equilibrium constant (K) and $k_f$ is the rate coefficient for the corresponding association reaction (given in Table 17). K determined from a modified van't Hoff plot over the temperature range 280-320 K using reported values of $\Delta H°_{298K}$ and $\Delta S°_{298K}$; [e] Based on consensus values of $\Delta H°_{298K}$ = -76.5 kJ mol$^{-1}$ and $\Delta S°_{298K}$ = -124.0 J mol$^{-1}$ K$^{-1}$ reported for the CH$_2$CHCH$_2$ + O$_2$ system by Rissanen et al. (2012), also taking account of the results of Ruiz et al. (1981), Morgan et al. (1982) and Knyazev and Slagle (1998); [f] K for the CH$_3$CHCHCH$_2$ + O$_2$ system calculated using $\Delta H°_{298K}$ = -81.05 kJ mol$^{-1}$ and $\Delta S°_{298K}$ = -132.25 J mol$^{-1}$ K$^{-1}$, being the average of values reported for the cis-CH$_3$CHCHCH$_2$ + O$_2$ and trans-CH$_3$CHCHCH$_2$ + O$_2$ systems by Knyazev and Slagle (1998). K assumed to be made up of a linear combination of 1/1.3 "-C=C-ĊHR + O$_2$" and 0.3/1.3 "-C=C-ĊH$_2$ + O$_2$", based on the relative importance of the association reactions (Table 17); [g] Assumed equivalent to rate coefficient for -C=C-CH(O$_2$)R → -C=C-ĊHR + O$_2$; [h] Based on a geometric average of rate coefficients calculated for (or assigned to) relevant peroxy radical structures formed from addition of OH and O$_2$ to isoprene. As recommended by Peeters (2015), these are based on those calculated by Peeters et al. (2014), but with each increased by a factor of 5 on the basis of the experimental characterization of the equilibration of peroxy radicals in each subset, as reported in preliminary form by Crounse et al. (2014) and applied in MCM v3.3.1; see Table S2 for further details; [i] Based on rate coefficients for E-1-OH-4-OO → trans-1-OH + O$_2$ and E-4-OH-1-OO → trans-4-OH + O$_2$ (see Table S2) , and also applied to corresponding secondary and tertiary radicals in the absence of data; [j] Based on rate coefficients for Z-1-OH-4-OO → cis-1-OH + O$_2$ and Z-4-OH-1-OO → cis-4-OH + O$_2$ (see Table S2) , and also applied to corresponding secondary and tertiary radicals in the absence of data; [k] Based on rate coefficients for 4-OH-3-OO → cis-4-OH + O$_2$ and 4-OH-3-OO → trans-4-OH + O$_2$ (see Table S2); [l] Based on rate coefficients for 1-OH-2-OO → cis-1-OH + O$_2$ and 1-OH-2-OO → trans-1-OH + O$_2$ (see Table S2).