# Peer review of "Estimation of rate coefficients and branching ratios for gas-phase reactions of OH with aliphatic organic compounds for use in automated mechanism construction"

_Atmospheric Chemistry and Physics, 2018_

## Referee Comment (RC1) · Anonymous Referee #1 · 28 Mar 2018

General Comments

This paper describes structure-activity relationship (SAR) methods for the gas-phase reactions of the OH radical with a large number of aliphatic hydrocarbons and oxygenated organic compounds. The approach involves determination of a partial rate coefficient for each reactive site in the compounds, thus enabling overall rate coefficients and branching ratios to be obtained. It successfully builds on previous efforts in the area by updating existing parameters for common compounds, while also extending the approach to new groups of compounds. The scope of the paper is wide-ranging and

considers virtually all types of aliphatic hyrdocarbons, monofunctional and bifunctional oxygenated organic compounds. This paper is part of a series by the same authors - SAR methods for the reactions of aromatic organic compounds are considered in a companion paper, with the reactions of peroxy radicals being covered in another.

Overall, this paper is very carefully written and remarkably error-free. The large amounts of information contained in it are presented in a logical and neat fashion. In particular, the supplementary information will be a very valuable resource for the atmospheric chemistry community. The research is well within the scope of Atmospheric Chemistry and Physics: I recommend publication after the authors have addressed the comments below.

Specific Comments:

1. Figures 1-7: There are a number of significant outliers in the log-log correlation plots. It would be useful to mention which compounds these are and if possible discuss why they are outliers.

2. Page 8, lines 22-24: Here, I think it would be worth referring to the work of Porter et al. (1997) who first highlighted the inability of Kwok and Atkinson's SAR to predict the reactivity of diethers and proposed that some type of Hydrogen-bonding like interaction between OH and the ethers is taking place. This idea was further discussed by Smith and Ravishankara (2002), who also considered hydrogen bonding for OH reactions with a wider range of oxygenated organic compounds.

3. Page 19, Section 6: Up until this point in the paper, the emphasis has been solely on estimation of rate coefficients and branching ratios for OH attack at different sites in molecules. It therefore seems a little strange to include this section on the reactions of alkyl radicals (other than reaction with molecular oxygen to produce organic peroxy radicals). I think the authors should either justify the inclusion of this material here or leave it out and consider incorporating it into another of the planned papers in the series.

4. Page 24, Section 7: It might be useful to briefly comment on whether developing SARs for other organic compounds, e.g. halocarbons, Sulfur- and Nitrogen-containing compounds etc., would be worthwhile.

Technical Corrections:

1. Page 5, line 1, the rate coefficient symbol should be in italics.

2. Page 7, lines 6-8, the authors should refer to Figure 2 here.

3. Page 10, line 34: Nielsen et al. (1989) is not included in the list of references.

4. Page 11, line 1: Liu et al. (1990) is not included in the list of references.

5. Page 16, line 3: Typo – should be Table 10.

6. Page 18, line 28: Boodaghians et al. (1989) is not included in the list of references.

7. Page 31-38: The following phrase contained in the figure captions of the log-log plots " the $\pm$ a factor of 2 range" is a bit awkward and should be reworded.

References:

Porter, E., Wenger, J., Treacy, J., Sidebottom, H., Mellouki, A., Téton S., and LeBras, G.: Kinetic Studies on the Reaction of Hydroxyl Radicals with Diethers and Hydroxy Ethers, J. Phys. Chem. A, 101, 5770-5775, 1997.

Smith, I.W.M., and Ravishankara, A.R.: Role of Hydrogen-Bonded Intermediates in the Bimolecular Reactions of the Hydroxyl Radical, J. Phys. Chem. A, 106, 4798–4807, 2002.
* * *

---

## Referee Comment (RC2) · G.S. Tyndall (Referee) · 18 Apr 2018

This paper explores a new Structure-Activity Relationship (SAR) for H-atom abstraction by OH, and for OH-addition to a wide number of organic compounds. The work more than builds on previous studies. It is thorough and wide-reaching, and treats different classes of molecules consistently. The paper is presented clearly and logically, so it is easy to follow the arguments made in building up the SAR. I tried calculating a few rate constants for representative molecules in the supplement, and they worked!

One minor issue is that the increase in scope of the study comes with an increase in complexity of the SAR. There are now many more k's and F's than previously, and while the SAR is clearly going to be of value for automated mechanism generation, it might be a little daunting for every day users who want to quickly estimate a few rate constants.

For example, I noted around 10 different rate constant representations for OH + aldehydes in Table 4. This is presumably required to explain the observed differences in reactivity, but more fundamentally may represent a limit of the SAR concept where negative activation energies are involved (aldehydes, multifunctional oxygenates).

Maybe a little more discussion could be given to classes of compounds that are not represented very well, as a caveat to users. In addition to those mentioned above, I note that rate constants for alkanes with a lot of tertiary hydrogens are often overestimated, too. Steric effects coming into play? This also has ramifications for estimating the site of attack, too.

I have only minor comments on the manuscript, which is well prepared, and clear, as noted above.

Page 6, line 17. Regarding the inductive effect on OH + aldehydes: While outside the scope of this work, it is well known that the presence of halogens deactivates the aldehyde group strongly.

Section 4.1. Maybe explain explicitly that in contrast to the method of Kwok and Atkinson, this follows Peeters, in considering each end of the double bond separately (just to avoid any confusion). I like the Peeters approach, since it allows a better estimate of the potential product distribution if the alkoxy radicals formed have more than one decomposition pathway.

Table 14 (and text). These are described as "prompt" reactions of thermalized radicals, but I usually associate that term with excited (i.e. nascent) radicals, not stabilized ones.

P13 line 1: Should be Table 10.

Page 17, line 29-30: "formation of a resonant radical is not represented" is not totally clear. I assume you mean that the unpaired electron on the formyl group is perpendicular to the orbitals of the C=C bond and cannot overlap. Represented may not be the right word though.

Page 26, line 15. Noziere accent misplaced.

Page 29, line 6. Typo Tyndal should be Tyndall.

Table 4: Sixth entry has pentavalent carbon R = –CH(X)(Y)Z. Also, Arrhenius expression in Table 4 header has missing (or extra) parenthesis.

The representation of different functional groups in the tables of F(X) values is a little confusing at first. I take it that you are only showing the relevant substituents that affect the rate constant. So in Table 12, for example, under column X we see –C=CH-C=CHR and the third carbon has only 3 bonds. It took a little while to figure out the logic, but it works (I think).

Overall, this is a very impressive piece of work that will be of great value to the community.

---

## Author Response (AR1)

**This file contains:**

- **Authors' responses to referee and discussion comments on:** Jenkin et al., Atmos. Chem. Phys. Discuss., https://doi.org/10.5194/acp-2018-145, 2018.

  Point-by-point responses to the review comments (original comments are shown in blue font). These responses also include a list of relevant changes made in the manuscript (shown in red font).

- **A marked-up version of the revised manuscript**.
* * *
**A. Comments by Referee 1**

**General comments**:

This paper describes structure-activity relationship (SAR) methods for the gas-phase reactions of the OH radical with a large number of aliphatic hydrocarbons and oxygenated organic compounds. The approach involves determination of a partial rate coefficient for each reactive site in the compounds, thus enabling overall rate coefficients and branching ratios to be obtained. It successfully builds on previous efforts in the area by updating existing parameters for common compounds, while also extending the approach to new groups of compounds. The scope of the paper is wide-ranging and considers virtually all types of aliphatic hyrdocarbons, monofunctional and bifunctional oxygenated organic compounds. This paper is part of a series by the same authors - SAR methods for the reactions of aromatic organic compounds are considered in a companion paper, with the reactions of peroxy radicals being covered in another.

Overall, this paper is very carefully written and remarkably error-free. The large amounts of information contained in it are presented in a logical and neat fashion. In particular, the supplementary information will be a very valuable resource for the atmospheric chemistry community. The research is well within the scope of Atmospheric Chemistry and Physics: I recommend publication after the authors have addressed the comments below.

Response: We are very grateful to the referee for these very positive and supportive comments on our work.

**Comment A1**: Figures 1-7: There are a number of significant outliers in the log-log correlation plots. It would be useful to mention which compounds these are and if possible discuss why they are outliers.

Response: We agree that it is important for the reader to know for which compounds the methods do not work so well, and aimed to provide information on this in the relevant sections. For clarity of presentation, we generally decided not to identify outliers on the log-log plots themselves, but to highlight and discuss relevant specific compounds or compound classes in the text. In practice there are only 22 outliers (defined as where the 298 K values of $k_{calc}$ and $k_{obs}$ differ by more than a factor of 2) out of the 486 points, as listed in the table below. The final column of the table identifies where additional information or discussion is given, this generally being for classes of compound where there are more than one isolated outlier (and also often covering systematic tendencies towards over- and underestimation when the discrepancy is less than a factor of two). For example, in Sect. 3.2.1, the statement "the method systematically underestimates the rate coefficients for 1,3- and 1,4-di-alcohols, by factors in the range 1.7 - 2.5" appears, with related discussion; and in Sect. 3.2.5 the statement "the results indicate that the rate coefficients for acyclic dinitrates and hydroxynitrates are apparently systematically underestimated, whereas those for the cyclic compounds tend to be overestimated", with related discussion.

We previously judged that our approach was sufficient, because observed and calculated values of the rate coefficient in the preferred data in the Supplement allow the reader to find the specific identities of the outliers. In view of the referee's comment, however, we realise that that this is not as clear and straightforward as it could be. We have therefore included the table below (as Table S1) in the Supplement of the revised paper, with this referred to in the captions of the relevant figures (see also the opening comment of the review by G.S. Tyndall).

**Identities of outliers, where the 298 K values of $k_{calc}$ and $k_{obs}$ differ by more than a factor of two.**

| Systematic name | Common name | $k_{calc}/k_{obs}$ | Compound class [a] |
|---|---|---|---|
| tetracyclo[2.2.1.0$^{2,6}$.0$^{3,5}$]heptane | quadricyclane | 0.469 | poly-cycloalkanes (Fig. 1) |
| butan-1,4-diol | | 0.364 | di-alcohols (Fig. 2, Sect. 3.2.1) |
| 6,6-dimethyl-bicyclo[3.1.1]heptan-2-one | nopinone | 0.257 | cycloketones (Fig. 2, Sect. 3.2.1) |
| [2,2-dimethyl-3-(2-oxo-propyl)-cyclopropyl]-acetaldehyde | caronaldehyde | 0.474 | keto-/di-aldehydes (Fig. 2, Sect. 3.2.1) |
| di-i-propoxy-methane | | 0.490 | acyclic diethers (Fig. 3, Sect. 3.2.3) |
| methanoic acid 2-methanalyloxy-ethyl ester | ethylene glycol diformate | 3.98 | di-esters (Fig. 4, Sect. 3.2.4) |
| ethanoic acid 2-ethanalyloxy-ethyl ester | ethylene glycol diacetate | 2.16 | di-esters (Fig. 4, Sect. 3.2.4) |
| pentandioic acid dimethyl ester | dimethyl glutarate | 2.17 | di-esters (Fig. 4, Sect. 3.2.4) |
| 1,2-dinitrooxybutane | | 0.361 | acyclic alkyl dinitrates (Fig. 5, Sect. 3.2.5) |
| 2,3-dinitrooxybutane | | 0.246 | acyclic alkyl dinitrates (Fig. 5, Sect. 3.2.5) |
| 1,4-dinitrooxybutane | | 0.165 | acyclic alkyl dinitrates (Fig. 5, Sect. 3.2.5) |
| 2-nitrooxy-1-propanol | | 0.264 | acyclic hydroxynitrates (Fig. 5, Sect. 3.2.5) |
| 2-nitrooxy-1-butanol | | 0.323 | acyclic hydroxynitrates (Fig. 5, Sect. 3.2.5) |
| 3-nitrooxy-1-butanol | | 0.416 | acyclic hydroxynitrates (Fig. 5, Sect. 3.2.5) |
| 5-nitrooxy-2-pentanol | | 0.344 | acyclic hydroxynitrates (Fig. 5, Sect. 3.2.5) |
| 4-nitrooxy-1-pentanol | | 0.259 | acyclic hydroxynitrates (Fig. 5, Sect. 3.2.5) |
| 6-nitrooxy-1-hexanol | | 0.324 | acyclic hydroxynitrates (Fig. 5, Sect. 3.2.5) |
| 3-methyl-cyclohex-2-enone | | 2.53 | $\alpha,\beta$-unsaturated ketones (Fig. 7, Table 11) |
| 3,5,5-trimethyl-cyclohex-2-enone | | 3.15 | $\alpha,\beta$-unsaturated ketones (Fig. 7, Table 11) |
| cis-3-hexene-2,5-dione | | 0.442 | $\alpha,\beta$-unsaturated ketones (Fig. 7, Table 11) |
| 3,4-dihydroxy-3-hexene-2,5-dione | | 0.322 | $\alpha,\beta$-unsaturated keto-enols (Fig. 7, Table 11) |
| 4-nitrooxy-but-2-en-1-ol | | 2.83 | hydroxyalkenyl nitrates (Fig. 7) |

**Comments:** [a] Class assigned to given compound in the identified figure. Where relevant, additional information and/or discussion is provided in the section or table shown.

**Comment A2**: Page 8, lines 22-24: Here, I think it would be worth referring to the work of Porter et al. (1997) who first highlighted the inability of Kwok and Atkinson's SAR to predict the reactivity of diethers and proposed that some type of Hydrogen-bonding like interaction between OH and the ethers is taking place. This idea was further discussed by Smith and Ravishankara (2002), who also considered hydrogen bonding for OH reactions with a wider range of oxygenated organic compounds.

Response: We agree that these highly-relevant papers should have been referred to, and this oversight has been corrected in the revised manuscript. The work of Porter et al. (1997) is now referred to in Sect. 3.2.3, where the following text appears (new text in red font):

"Both studies report difficulties in recreating the rate coefficients for the complete series of compounds, with discrepancies of up to over a factor of three between estimated and observed values. This was also considered previously in the work of Porter et al. (1997), who proposed that these deviations may be a consequence of stabilization of the reaction transition states by hydrogen bonding."

The work of both Porter et al. (1997) and Smith and Ravishankara (2002) is now also referred to in the Conclusions (Sect. 7), where general statements are made about the performance of the SAR for oxygenated organic compounds (see also the opening comment of the review by G.S. Tyndall):

"Fig. 11 also shows that the reliability of the SARs decreases with the number of functional groups on the carbon skeleton. Indeed, the RMSE increases from 0.07 for hydrocarbons to 0.13 for monofunctional species and reaches 0.22 for multifunctional species, i.e. a relative error for the calculated $k_{298K}$ of 17 %, 35 % and 66 %, respectively. This reflects the effects of the presence of polar oxygenated functional groups, and difficulties in accounting fully for their long-range influences through stabilization of transition states by hydrogen bonding (e.g. Porter et al., 1997; Smith and Ravishankara, 2002; Calvert et al., 2011)."

**Comment A3**: Page 19, Section 6: Up until this point in the paper, the emphasis has been solely on estimation of rate coefficients and branching ratios for OH attack at different sites in molecules. It therefore seems a little strange to include this section on the reactions of alkyl radicals (other than reaction with molecular oxygen to produce organic peroxy radicals). I think the authors should either justify the inclusion of this material here or leave it out and consider incorporating it into another of the planned papers in the series.

Response: As the referee indicates, it is necessary to include information on how the (usually rapid) reactions of the product organic radicals are treated somewhere in the proposed set of papers. We judged that there is insufficient information to justify a standalone paper on this topic, so we had to make a decision on which of the other proposed papers would be the most appropriate home for this information. After some consideration, we decided that this paper (and the companion paper) treating the OH initiation reactions were the most appropriate. This decision is now justified.

Mechanisms used in atmospheric models invariably represent the rapid reactions of product radicals as part of the initiation reaction, with this most commonly being reaction with $O_2$ to form a thermalized peroxy radical, e.g. in the case of the reaction of OH with methane:

$OH + CH_4 (+ O_2) = CH_3O_2 (+ H_2O)$

where the species in brackets are not usually represented in atmospheric mechanisms, but are shown here to balance the equation. It is therefore clearly valid to make this point in the present paper (as the referee also states), and this is therefore covered in the introduction to Sect. 6, and in Sect. 6.3 for resonant radicals.

In some cases, however, the product radical may undergo a very rapid decomposition or rearrangement to form a different radical product, with this process being represented as part of the initiation reaction, e.g. in the case of H atom abstraction from the $CH_3$ group in methyl hydroperoxide, the initiation reaction would usually be represented as follows:

$OH + CH_3OOH = HCHO + OH (+ H_2O)$

In other cases, the initial adduct from the reaction of the product radical with $O_2$ may not form a thermalized peroxy radical, but rearrange and decompose to form a different radical product - with this process being represented as part of the initiation reaction; e.g. in the case of H atom abstraction from the $CH_3$ group in methanol, the initiation reaction would usually be represented as follows:

$OH + CH_3OH (+ O_2) = HCHO + HO_2 (+ H_2O)$

It is clearly equally valid to make these points in the present paper, which are covered by the information in Sects. 6.1 and 6.2, respectively. A main aim of Sect. 6 is therefore to summarize information of this type to help define which products are represented in a mechanism following the reactions of OH with the organic compounds discussed in the earlier sections. In our view, this is completely justifiable – and totally analogous (for example) to presenting information on the reactions of Criegee intermediates in a paper on the reactions of $O_3$ with alkenes.

Where we have sympathy with the referee's point is that we decided it was also necessary to include information for radicals formed from other sources, the fates of which sometimes differ from those of the same radicals formed from the OH reactions, owing to a lack of chemical activation. While this information is not so directly relevant to the present paper, we feel it would be artificial and confusing to separate it out, such that Sect. 6 therefore provides general information on the reactions of organic radicals that are generally not represented explicitly in chemical mechanisms.

**Comment A4**: Page 24, Section 7: It might be useful to briefly comment on whether developing SARs for other organic compounds, e.g. halocarbons, Sulfur- and Nitrogen-containing compounds etc., would be worthwhile.

Response: The referee raises a very sensible point. To make the task manageable, we have limited our work to reactions relevant to the degradation of hydrocarbons and oxygenates. However, we agree that the chemistry of emitted compounds containing halogens, sulphur and nitrogen needs to

be kept under review. The following new text now appears at the end of Sect. 7 in the revised manuscript:

"This work has focused on the reactions of OH radicals with hydrocarbons and oxygenated organic compounds, which play a central role in tropospheric chemistry. Although outside the scope of the present study, it is noted that development of SAR methods for reactions with emitted organic compounds containing halogens, sulphur and nitrogen would also be of value."

**Technical comments, typos and additional references**:

Page 5, line 1, the rate coefficient symbol should be in italics.

Page 7, lines 6-8, the authors should refer to Figure 2 here.

Page 10, line 34: Nielsen et al. (1989) is not included in the list of references.

Page 11, line 1: Liu et al. (1990) is not included in the list of references.

Page 16, line 3: Typo – should be Table 10.

Page 18, line 28: Boodaghians et al. (1989) is not included in the list of references.

Response: We are very grateful to the referee for identifying the above typos and missing references, which have been corrected in the revised manuscript.

Page 31-38: The following phrase contained in the figure captions of the log-log plots " the ± a factor of 2 range" is a bit awkward and should be reworded.

Response: This is a fair point. In each case, we have changed the relevant phrase to read more simply "The broken lines show the factor of 2 range".

**B. Comments by G. S. Tyndall (Referee)**

**Opening comment:**

This paper explores a new Structure-Activity Relationship (SAR) for H-atom abstraction by OH, and for OH-addition to a wide number of organic compounds. The work more than builds on previous studies. It is thorough and wide-reaching, and treats different classes of molecules consistently. The paper is presented clearly and logically, so it is easy to follow the arguments made in building up the SAR. I tried calculating a few rate constants for representative molecules in the supplement, and they worked!

One minor issue is that the increase in scope of the study comes with an increase in complexity of the SAR. There are now many more k's and F's than previously, and while the SAR is clearly going to be of value for automated mechanism generation, it might be a little daunting for every day users who want to quickly estimate a few rate constants.

For example, I noted around 10 different rate constant representations for OH + aldehydes in Table 4. This is presumably required to explain the observed differences in reactivity, but more fundamentally may represent a limit of the SAR concept where negative activation energies are involved (aldehydes, multifunctional oxygenates).

Maybe a little more discussion could be given to classes of compounds that are not represented very well, as a caveat to users. In addition to those mentioned above, I note that rate constants for alkanes with a lot of tertiary hydrogens are often overestimated, too. Steric effects coming into play? This also has ramifications for estimating the site of attack, too.

I have only minor comments on the manuscript, which is well prepared, and clear, as noted above.

Response: We are very grateful to the referee for the positive and supportive comments on the methods and manuscript, and for the additional comments and observations.

As presented and discussed in the manuscript (mainly in Sects. 3.2.1 and 3.2.3), the complexity of the SAR has increased for some compound classes to allow improvements in performance compared with earlier methods. Table 4 actually contains seven generic rate coefficients (i.e. those with

parameter names) describing the reactivity of the formyl group in aldehydes possessing different classes of organic group (as stated in Sect. 3.2.1 and the notes to Table 4, the other three un-named rate coefficients are specific to unique compounds for which measured rate coefficients are available, and are shown only to illustrate the trend in increasing substitution in R). Although this necessarily increases the complexity of the method, we decided this approach is less confusing than the alternative of defining a single rate coefficient with a set of seven substituent factors that are specific to formyl hydrogen abstraction, and very different from those for abstraction from other C-H bonds (e.g. as appearing in Table 5).

We agree that the method adjustments reflect difficulties in defining SAR methods for oxygenated organic compounds, which likely result at least partially from the role of hydrogen bonding in reaction transition states. As indicated above in the response to Comment A2 by Referee 1, we have now included some additional discussion of this point, with reference to key publications.

The referee raises an interesting point about abstraction of tertiary hydrogens and the possible impact of steric hindrance, and we have looked into this further. The table below shows the observed and calculated 298 K rate coefficients for the ten acyclic alkanes containing tertiary hydrogens in the database. Although $k_{calc}$ for the alkane containing the most tertiary hydrogens (2,3,4-trimethylpentane) is overestimated by about 15 % (possibly consistent with a small steric effect) there is no clear trend with increasing numbers of tertiary hydrogens. Consideration of the data for 2,2,3-trimethylbutane also suggests no impact of steric hindrance from the bulky -$C(CH_3)_3$ group adjacent to the tertiary hydrogen, with $k_{calc}$ underestimating the reactivity. This may therefore actually hint that F(X) for -$CR_3$ groups should be greater than the value of 1.35 optimized for alkyl groups in general. In practice, we feel that additional data on a broader range of branched alkanes are required before such discrimination of F(X) values for different alkyl groups and establishment of steric hindrance effects are possible.

Concerning the more general point about discussing compound classes that are less well represented, we agree that more information was required. As indicated above in the response to Comment A1 by Referee 1, we have included more information on the identities of outliers in the revised manuscript, with clearer direction to where to find the relevant discussion in the text. As also indicated above, this has been further strengthened by specific discussion of the role of hydrogen bonding in reaction transition states for organic oxygenates, with reference to key publications.
* * *
**Comparison of the 298 K values of $k_{calc}$ and $k_{obs}$ for alkanes containing tertiary hydrogens**

|  | $k_{obs}$ | $k_{calc}$ | $k_{calc}/k_{obs}$ |
|---|---|---|---|
| *1 tertiary hydrogen* |  |  |  |
| 2-methylpropane | 2.10E-12 | 2.02E-12 | 0.960 |
| 2-methylbutane | 3.75E-12 | 3.58E-12 | 0.954 |
| 2-methylpentane | 5.26E-12 | 4.98E-12 | 0.946 |
| 3-methylpentane | 5.53E-12 | 5.32E-12 | 0.962 |
| 2,2,3-trimethylbutane | 3.81E-12 | 2.89E-12 | 0.758 |
| 2,2,4-trimethylpentane | 3.34E-12 | 4.29E-12 | 1.285 |
|  |  |  | 0.978 ± 0.170 (average) |
| *2 tertiary hydrogens* |  |  |  |
| 2,3-dimethylbutane | 6.14E-12 | 4.73E-12 | 0.770 |
| 2,3-dimethylpentane | 6.58E-12 | 6.47E-12 | 0.983 |
| 2,4-dimethylpentane | 5.70E-12 | 6.13E-12 | 1.075 |
|  |  |  | 0.943 ± 0.156 (average) |
| *3 tertiary hydrogens* |  |  |  |
| 2,3,4-trimethylpentane | 6.60E-12 | 7.62E-12 | 1.154 |

**Comment B1**: Page 6, line 17. Regarding the inductive effect on OH + aldehydes: While outside the scope of this work, it is well known that the presence of halogens deactivates the aldehyde group strongly.

Response: We agree that this is a very relevant observation, which is now made at the appropriate point in Sect. 3.2.1 in the revised manuscript.

"In conjunction with the observed increasing trend in $k_{298K}$ with increasing alkyl substitution in the organic group, it appears that the reactivity of the formyl group is influenced by the inductive effect of the organic group. Although outside the scope of the present study, it is well known that the inductive effect of halogens strongly deactivates the OH reactivity of the formyl group in halogen-substituted aldehydes (e.g. Scollard et al., 1993)."

**Comment B2**: Section 4.1. Maybe explain explicitly that in contrast to the method of Kwok and Atkinson, this follows Peeters, in considering each end of the double bond separately (just to avoid any confusion). I like the Peeters approach, since it allows a better estimate of the potential product distribution if the alkoxy radicals formed have more than one decomposition pathway.

Response: We thank the referee for this suggestion and endorsement. In addition to the existing detailed description of the method in the ensuing subsections, we have now included this point in the Sect. 4.1 introduction, as follows:

"The estimation of rate coefficients for OH addition to C=C bonds ($k_{add}$) is based on the method described by Peeters et al. (2007), but is extended to include the effects of hydrocarbon and oxygenated substituent groups. In contrast to the earlier SAR methods (e.g. Kwok and Atkinson, 1995), the Peeters et al. (2007) approach represents addition of OH to either end of the C=C bond explicitly, and therefore allows the attack distribution to be defined."

**Comment B3**: Table 14 (and text). These are described as "prompt" reactions of thermalized radicals, but I usually associate that term with excited (i.e. nascent) radicals, not stabilized ones.

Response: This is a fair point. The word "prompt" has been replaced by "rapid" in the relevant text in Sect. 6.1 and in the Table 14 caption.

**Comment B4**: P13 line 1: Should be Table 10.

Response: We are grateful to the referee for identifying this typo, which has been corrected in the revised manuscript.

**Comment B5**: Page 17, line 29-30: "formation of a resonant radical is not represented" is not totally clear. I assume you mean that the unpaired electron on the formyl group is perpendicular to the orbitals of the C=C bond and cannot overlap. Represented may not be the right word though.

Response: The referee is correct. The point being made was that it is not necessary to represent a resonant radical in this case, for the reason given by the referee. We agree that this was badly worded, and has been amended in the revised text as suggested:

"In the specific case of H-atom abstraction from a formyl group adjacent to a C=C bond, formation of a resonant radical is not possible (owing to the perpendicular alignment of the unpaired electron)"…."

**Comment B6**: Page 26, line 15. Noziere accent misplaced.

**Comment B7**: Page 29, line 6. Typo Tyndal should be Tyndall.

Response: We are grateful to the referee for identifying these typos, which have been corrected in the revised manuscript.

**Comment B8**: Table 4: Sixth entry has pentavalent carbon R = –CH(X)(Y)Z. Also, Arrhenius expression in Table 4 header has missing (or extra) parenthesis.

Response: We are grateful to the referee for identifying the valency error in Table 4, which has been corrected to "-C(X)(Y)Z" in the revised manuscript. The missing closing parenthesis has also been corrected in this, and other, figure captions.

**Comment B9**: The representation of different functional groups in the tables of F(X) values is a little confusing at first. I take it that you are only showing the relevant substituents that affect the rate constant. So in Table 12, for example, under column X we see –C=CH-C=CHR and the third carbon has only 3 bonds. It took a little while to figure out the logic, but it works (I think).

Response: The referee is correct that the identities of the substituents, "X", in the F(X) tabulations identify the functional groups that influence the rate coefficient. In most cases, these are shown explicitly, with full residual substitution by either hydrogen atoms (H), alkyl groups (R) or the relevant oxygenated group(s). In the specific case of Table 12, showing the effects of resonant groups with up to $C_4$ chains, substituents are only shown on the carbon radical centres that influence the value of F(X). The unrepresented substituents on the intermediate carbon atoms could be either H or R, but have no effect on the parameter. If these were represented, the structures would tend to become unwieldy and several structures would need to be shown for each F(X) value. For example, "-C=CH-C=CHR" would become:

"-CH=CHCH=CHR, -CH=CHC(R)=CHR, -C(R)=CHCH=CHR, -C(R)=CHC(R)=CHR".

The key points are that the second carbon in the $C_4$ chain is substituted by "H" and the fourth by "R" and "H", such that the resonant radical centres each have secondary character in this example. We feel the current representation shows this more clearly than including all the possible structures, and it is gratifying that the referee found it logical, if not immediately clear. In view of the referee's comment, the following text has been added to footnote "a" in Table 12: "For clarity, residual substituents are not shown on intermediate carbon atoms, but can be either H or R."

**Comment B10**: Overall, this is a very impressive piece of work that will be of great value to the community.

Response: Once again, we thank G. S. Tyndall for the supportive comments on our work, and for the clear and helpful suggestions for amendments and improvements.

[revised manuscript text omitted]

25 Atkinson, 1995), the Peeters et al. (2007) approach represents addition of OH to either end of the C=C bond explicitly, and therefore allows the attack distribution to be defined.

**4.1 Alkenes and polyalkenes**

**4.1.1 Acyclic monoalkenes**

For isolated C=C bonds in monoalkenes and polyalkenes, the Peeters et al. (2007) method defines site-specific parameters for

30 addition to form primary, secondary and tertiary β-hydroxyalkyl radicals, as follows:

$k(\text{-C=CH}_2) = k_{\text{prim-add}}$ $\qquad\qquad$ (8)

$k(\text{-C=CH-X}) = k_{\text{sec-add}}\ \text{F'(X)}$ $\qquad\qquad$ (9)

$k(\text{-C=C(-X)-Y}) = k_{\text{tert-add}}\ \text{F'(X)}\ \text{F'(Y)}$ $\qquad\qquad$ (10)

[revised manuscript text omitted]
 possible (owing to the perpendicular alignment of the unpaired electron), and a single rate coefficient ($k_{\text{abs(-CHO)-}\alpha\text{C=C}}$) was simultaneously optimized, based on data for 13 $\alpha,\beta$-unsaturated aldehydes (see Table 4). As with the other formyl group rate coefficients in Table 4, this rate coefficient is applied independently of substituent factors. A correlation of the resultant values of $k_{\text{calc}}$ with $k_{\text{obs}}$ at 298 K is shown in the upper panel of Fig. 7.

Temperature-dependent recommendations are available for a subset of 22 unsaturated organic oxygenates. Where possible, these were used to provide representative values of the temperature coefficients ($B_{\text{F(X)}}$) and pre-exponential factors ($A_{\text{F(X)}}$) for the substituent factors given in Table 2. The values of $B_{\text{F(X)}}$ were varied with the aim of minimizing the summed square deviation in the composite temperature coefficients, $\Sigma((E/R)_{\text{calc}}-(E/R)_{\text{obs}})^2$, for the contributing sets of compounds. The resultant $(E/R)_{\text{calc}}$ values are compared with the recommended $(E/R)_{\text{obs}}$ values in the lower panel of Fig. 7 (see also Fig. S9). The values of $A_{\text{F(X)}}$ were automatically returned from the corresponding optimized $B_{\text{F(X)}}$ and F(X)$_{298\text{K}}$ values. An optimized temperature-dependence expression for $k_{\text{abs(-CHO)-}\alpha\text{C=C}}$ was also determined as part of this procedure, as given in Table 4.

The site-specific partial rate coefficients estimated by the above SAR methods can also be used to define the branching ratios for both OH addition and H-atom abstraction for the reaction of OH with a given unsaturated oxygenate. Where available, the present methods appear to provide a reasonable representation of reported product yields and mechanistic information (e.g. see examples given in the Supplement).

**5 Unsaturated organic compounds containing C≡C bonds**

Reported kinetic data for the reactions of OH with alkynes are available for ethyne (acetylene), propyne, but-1-yne, but-2-yne, pent-1-yne and hex-1-yne. These data suggest that the rate coefficient for OH addition to C≡C bonds in alkynes cannot be estimated in an analogous way to that applied to alkenes above (i.e. by adding partial rate coefficients for addition of OH to each side of the triple bond). In this case, the addition rate coefficient is based on a single parameter for the C≡C group ($k_{\text{C≡C}}$), which is modified on the basis of the identities of the two substituent groups:

$$k(\text{X-C≡C-Y}) = k_{\text{C≡C}} \, \text{F}_{\text{C≡C}}(\text{X}) \, \text{F}_{\text{C≡C}}(\text{Y}) \tag{19}$$

The values of $k_{\text{C≡C}}$ and relevant $\text{F}_{\text{C≡C}}(\text{X})$ were optimized using the preferred dataset, with rate coefficients based on high pressure limiting values. For this procedure, $\text{F}_{\text{C≡C}}(\text{-H})$ was assigned a value of 1.00, and the abstraction of H atoms from the substituent alkyl groups was treated using the method optimized above in Sect. 3, with F(-C≡C-) also assumed to take a value of 1.00 (as previously applied by Kwok and Atkinson, 1995). This resulted in an optimized value of $k_{\text{C≡C}} = 9.4 \times 10^{-13}$ cm$^3$ molecule$^{-1}$ s$^{-1}$, with $\text{F}_{\text{C≡C}}(\text{-CH}_3) = 4.8$ for a methyl substituent, and $\text{F}_{\text{C≡C}}(\text{-R}) = 8.0$ applied to all other alkyl substituents (although the data are

[revised manuscript text omitted]

In some cases, organic radicals formed specifically from the reactions of OH with VOCs are formed chemically activated, $[R]^{\ddagger}$, and the rate of decomposition or rearrangement is enhanced. These are represented as follows:

(i) Abstraction of the formyl H atom in methylglyoxal ($CH_3C(=O)CHO$) via reaction with OH has been reported to generate activated $[CH_3C(=O)CO]^{\ddagger}$ radicals which decompose promptly and exclusively (Baeza-Romero et al., 2007). This is therefore also assumed for $[R'C(=O)CO]^{\ddagger}$ formed specifically from the reactions of OH with higher analogues (where R' is any organic group), leading to the following overall reaction:

$$OH + R'C(=O)CHO \qquad \rightarrow R' + CO + CO \; (+ H_2O) \; (40\text{ %}) \qquad\qquad\qquad\qquad (R2a)$$

$$\rightarrow R'\dot{C}=O + CO \; (+ H_2O) \;\; (60\text{ %}) \qquad\qquad\qquad\qquad\qquad (R2b)$$

[revised manuscript text omitted]
. This reflects the effects of the presence of polar oxygenated functional groups, and difficulties in accounting fully for their long-range influences through stabilization of transition states by hydrogen bonding (e.g. Porter et al., 1997; Smith and Ravishankara, 2002; Mellouki et al., 2003; Calvert et al., 2011). In the multifunctional subset (124 species), most of the species are bifunctional compounds (116 species), with a limited contribution from trifunctional compounds. The reliability of the SARs for species with more than two functional groups can therefore not be assessed. The atmospheric oxidation of hydrocarbons and organic oxygenates likely leads to a myriad of highly functionalized species (e.g. Aumont et al., 2005, 2012, Goldstein and Galbally, 2007; Mentel et al., 2015). Extrapolation of the SAR to this category of compound is therefore required in models aiming to describe atmospheric oxidation explicitly. Additional rate coefficients would therefore be highly valuable for further assessment and constraining of

SARs for multifunctional species. Finally, for the full database, the SARs give fairly reliable $k_{298K}$ estimates, with a MAE of 0.09 and a RMSE of 0.15, corresponding to an overall agreement of the calculated $k_{298K}$ within 40%.

This work has focused on the reactions of OH radicals with hydrocarbons and oxygenated organic compounds, which play a central role in tropospheric chemistry. Although outside the scope of the present study, it is noted that development of SAR methods for reactions with emitted organic compounds containing halogens, sulphur and nitrogen would also be of value.

**Acknowledgements**

This work received funding from the Alliance of Automobile Manufacturers, and as part of the MAGNIFY project, with funding from the French National Research Agency (ANR) under project ANR-14-CE01-0010, and the UK Natural Environment Research Council (NERC) via grant NE/M013448/1. It was also partially funded by the UK National Centre for Atmospheric Sciences (NCAS) Composition Directorate. Marie Camredon (LISA, Paris) and Luc Vereecken (Forschungszentrum Jülich) is are gratefully acknowledged for helpful discussions on this work. We also thank Geoff Tyndall (NCAR, Boulder) and an anonymous referee for review comments and suggestions that helped to improve the manuscript.

[revised manuscript text omitted]